



# Tropical temperature evolution across two glacial cycles derived from speleothem fluid inclusion microthermometry

Yves Krüger [1], Leonardo Pasqualetto [1], Alvaro Fernandez [3], Kim M. Cobb [4], A. Nele Meckler [1,2]

[1]Department of Earth Science, University of Bergen, Bergen, Norway

[2]Bjerknes Centre for Climate Research, University of Bergen, Bergen, Norway

[3]Andalusian Institute of Earth Sciences, CSIC-University of Granada, Granada, Spain

[4]Institute at Brown for Environment and Society, Department of Earth, Environmental & Planetary Sciences, Brown University, Providence, RI, USA

*Correspondence to*: Yves Krüger (yves.kruger@uib.no)

**Abstract.** The evolution of tropical temperature across multiple glacial-interglacial cycles is mostly constrained with marine proxy records, which are associated with considerable uncertainties. Here we present a reconstruction of tropical land temperatures derived from fluid inclusions in stalagmite WR5_B from Whiterock Cave (Gunung Mulu National Park, Northern Borneo). The employed paleothermometer - nucleation-assisted microthermometry - is based on the density of the
water trapped in fluid inclusions, i.e., on a well-known thermodynamic parameter and yields highly precise temperature estimates. The record consists of 49 temperature data points spanning a 127 kyr period from 460 to 333 ka including the glacial terminations T-V and T-IV. We find that Borneo temperature tracks Southern hemisphere temperature and atmospheric $CO_2$ concentrations. Deglacial warming is accompanied by relatively dry conditions in Northern Borneo, indicated by pronounced enrichments in calcite $\delta^{18}O_{cc}$ and reconstructed drip water $\delta^{18}O_{dw}$ values. The amplitude of glacial-
interglacial temperature changes amounts to $4.2 \pm 0.4$ °C (2SEM) between MIS 12 and the MIS 11 interglacial optimum and $4.3 \pm 0.4$ °C (2SEM) across T-IV. MIS 11 peak temperature was found to be $0.9 \pm 0.4$ °C warmer than late Holocene temperatures reconstructed for Whiterock Cave, whereas temperatures during MIS 12 and MIS 10 glacial maxima in our record are indistinguishable from those previously reconstructed for the Last Glacial Maximum. Both the present WR5_B record as well as the recently published record from Løland et al. (2022) covering the last glacial Termination exhibit a clear
linear correlation with Antarctic temperature anomalies ($R^2 = 0.89$ and $0.97$, respectively), with practically identical slopes of the linear regression lines. Depending on the employed Antarctic $\Delta T$ reconstruction, Landais et al. (2021) and Jouzel et al. (2007), we found a polar amplification factor of $2.21 \pm 0.22$ and $2.42 \pm 0.23$ (95% CI), respectively.



## 1 Introduction

The glacial-interglacial cycles of the Pleistocene offer valuable opportunities to study the response of the global climate
system to varying boundary conditions such as atmospheric $CO_2$ concentration, and to identify the processes that
characterize large global climate shifts between glacial and interglacial states. The larger glacial-interglacial cycles of the last
million years feature many similarities, but also clear differences, especially in the magnitude, duration, and temporal
evolution of the interglacials (Past Interglacial Working Group of PAGES, 2016). Improved constraints as to how these
interglacials differed in terms of global and regional characteristics will yield important insights into the interactions and
sensitivities of the different climate system components.

Quantitative temperature reconstructions are crucial to assessing the global and regional sensitivity to varying greenhouse
forcing and furthermore serve as important benchmarks for earth system models. However, apart from the detailed
temperature reconstructions from Antarctic ice cores (Jouzel et al., 2007; Landais et al., 2021), the vast majority of glacial-
interglacial temperature reconstructions come from marine sediments for long-term temperature records with scant records
derived from land. Marine temperature proxies are associated with uncertainties regarding the water depth and seasonality of
the signal production (Ho and Laepple, 2016, Bova et al., 2021) and are often affected by additional, non-thermal factors
such as salinity, pH and biological effects (e.g., Gray and Evans, 2019). Discrepancies between proxy-based temperature
reconstructions underly a long-standing debate on the amplitude of glacial-interglacial temperature changes in the tropics
(see an overview in Tripati et al., 2014).

The advancement of several quantitative proxy methods for speleothem-based temperature reconstructions such as $\delta^{18}O$
thermometry based on fluid inclusion and calcite oxygen isotopic compositions (e.g. Affolter et al., 2014), noble gas
thermometry (e.g. Kluge et al., 2008), clumped isotope thermometry (e.g. Affek et al. 2008) or TEX86 (e.g. Baker et al.,
2019) opens new opportunities to fill in the gap of terrestrial temperature data by producing quantitative records from
different karst areas around the globe. Here we employ what is arguably the most precise and accurate of these methods (*cf.*
Meckler et al., 2015), nucleation-assisted (NA) microthermometry, which is based on fluid inclusion liquid-vapour
homogenisation (Krüger et al., 2011). The main advantages of this paleothermometer are its physical basis (circumventing
the need for empirical calibration) and its high precision (typical 2 sigma standard errors of the mean (2SEM) are around 0.3
°C).

Here, we applied NA microthermometry to a stalagmite from Northern Borneo, located in the heart of the the Western
Pacific Warm Pool (Fig. 1). Stalagmite WR5_B covers almost two glacial-interglacial cycles from Marine Isotope Stage
(MIS) 12 to MIS 9, including two glacial terminations, T-V (also known as Mid-Brunhes transition) and T-IV, as well as the
glacial inception from MIS 11 to MIS 10. MIS 11 is an unusually long and warm interglacial period (Tzedakis et al., 2022;
Past Interglacial Working Group of PAGES, 2016) that, like the Holocene, is characterised by relatively weak insolation
forcing, i.e., a small amplitude of the precession cycle due to the low eccentricity of the Earth's orbit. Using the same



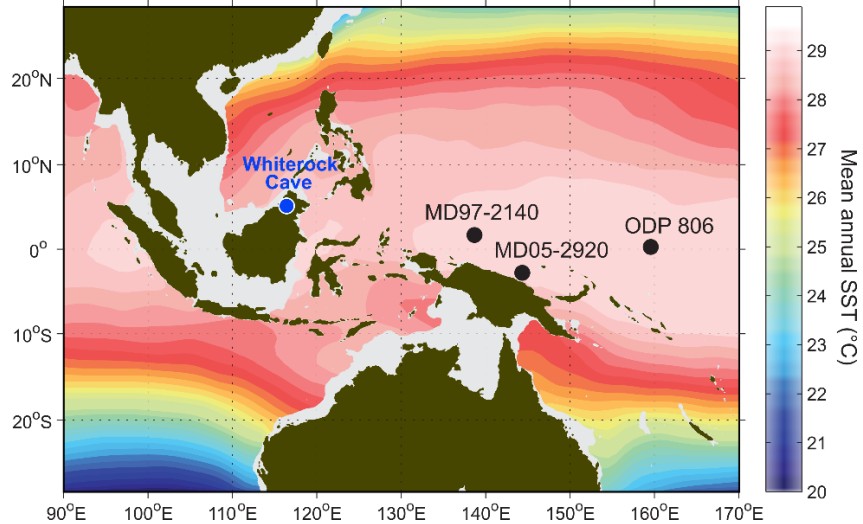

**Figure 1: Map of modern-day mean annual sea surface temperatures (SST) in the Western Pacific Warm Pool region. Shelf areas that were exposed during glacial sea level lowstands (-120 m) are shown in white. The locations of Whiterock Cave (Gunung Mulu National Park, Northern Borneo) as well as three marine core sites with SST reconstructions relevant to this study (see Section 4.2) are indicated with dots. Modified after Meckler et al. (2012; Supplementary Material)**

analytical method, Løland et al. (2022) presented a temperature reconstruction across the last glacial termination showing that Borneo land temperature closely tracks the deglacial evolution of atmospheric $CO_2$ and Antarctic temperature and is largely decoupled from hydroclimate changes that are strongly impacted by Northern Hemisphere cooling events. Here we investigate whether the same holds true also for earlier glacial terminations and across glacial-interglacial cycles and we assess the amplitude of glacial interglacial temperature changes. Given that this paleothermometer is not yet widely established, we include a detailed description of the method and discuss uncertainties related to measurements and the resulting cave temperature reconstructions in Section 2.

## 2 Materials and Methods

### 2.1 Stalagmite sample

Stalagmite WR5 was collected in 2008 from Whiterock Cave (Gunung Mulu National Park, Northern Borneo, Malaysia; 4.1°N, 114.9°E, about 180 m a.s.l.). Present-day air temperature in Whiterock Cave is 23.7 °C. Continuous temperature monitoring close to the location of WR5 over a 30-month period, from March 2018 until September 2020, revealed minor fluctuations of the cave air temperature of about ± 0.1 °C (total amplitude), indicating very stable temperature conditions in the cave throughout the year. For comparison, temperatures recorded at the nearby meteorological station at Mulu airport (24 m a.s.l.) between 2006 and 2012 yielded an average outside air temperature of 24.3 °C over the entire period with annual mean temperatures varying between 24.0 and 24.7 ˚C. The higher average temperatures measured at Mulu airport are



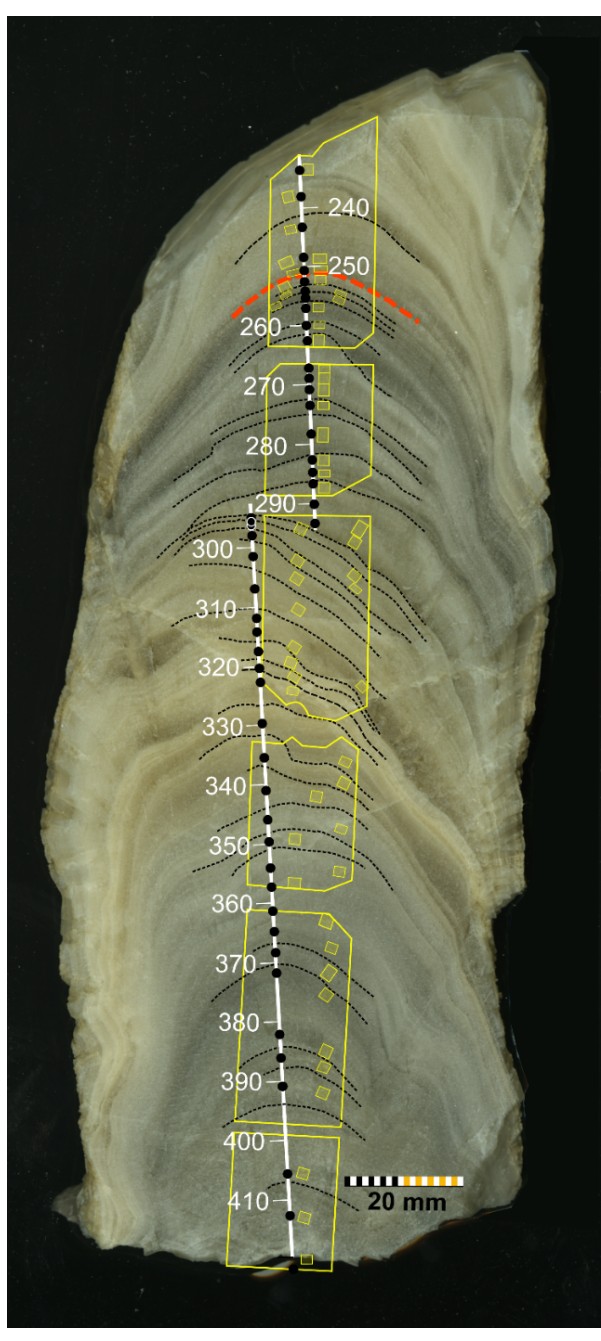

**Figure 2: Cross section of stalagmite WR5_B sub-parallel to the central growth axis. White lines with millimetre gradations indicate the location of the calcite isotope transect. Yellow polygons show the outlines of the sample blocks from which the thick sections for petrographic and fluid inclusion analysis were cut and their position with respect to the isotope transect. Yellow shaded squares label the 49 sample positions at which fluid inclusions were analysed for this study. Black dots represent the respective sample positions on the isotope transect. Dotted lines indicate growth hiatuses. Note, the red dotted line marks a major hiatus at the beginning of Termination IV (see Fig. A1a and h, and Fig. 7).**





generally consistent with the lower altitude of the meteorological station. Although the two temperature records do not cover the same time interval, it is safe to assume that present-day cave temperature in Whiterock Cave closely reflects (multi)-annual mean outside air temperature without seasonal bias.

WR5 is a fossil stalagmite with a total length of 400 mm that was found tipped over on a rock pile in a cave chamber about 350 m from the cave entrance. For the present study, we used the lower ca. 190 mm long section of the stalagmite denoted as WR5_B (Fig. 2) that covers a 127 kyr time interval from 460 to 333 ka, and thus, includes two glacial terminations (T-V and T-IV). WR5_B has previously been studied with regard to tropical hydroclimate using the oxygen isotope signal of the calcite, $\delta^{18}O_{cc}$ (Meckler *et al.*, 2012), and later on, was employed for a comparison study of different speleothem paleo-

thermometers (Meckler *et al.*, 2015).

WR5_B exhibits distinct colour changes that can be assigned to glacial (grey) and interglacial (yellow ochre) periods. The calcite fabric that was analysed on 200 μm thick polished stalagmite sections is predominantly columnar open (Frisia, 2015) showing large, up to cm-wide composite crystals that are build up from columnar crystal units of almost perfectly uniform *c*-axis orientation (Fig. A1a,b; see Appendix). Fluid inclusions of appropriate size ($10^3$-$10^5$ μm$^3$) are abundant and of primary

origin, i.e. they formed during crystal growth by incomplete lateral coalescence of adjacent columnar crystal units (*cf.* Kendall and Broughton, 1978). The uniform *c*-axis orientation of the columnar crystal units is likely to promote a tight sealing of the predominantly inter-crystalline inclusions. Characteristically, the inclusions are monophase liquid and exhibit elongate spindle-like shapes with a thorn-shaped tip at the upper end (Fig. A1c,d). Occasionally, two-phase inclusions are also present but these were ignored because their bulk water densities are not related to stalagmite formation temperatures.

They likely result from leakage or stretching of the inclusions, which at some point provokes a spontaneous nucleation of the vapour bubble.

Petrographic inspections of the thick sections allowed for identifying more than 30 potential growth interruptions in WR5_B. Some of these hiatuses display irregular surface structures (Fig. A1e) likely resulting from calcite dissolution (Type E surfaces; Railsback *et al.*, 2013), whereas others exhibit smooth surfaces marked by numerous tiny fluid inclusions (Fig.

A1f). After a hiatus, crystal growth generally continued with the same crystallographic orientation. In one case (at position 320 mm), however, small, potentially detrital calcite crystals of random crystallographic orientation can be observed on the hiatus surface (Fig. A1g). Subsequently, they were overgrown by the columnar calcite fabric due to geometric selection and competitive growth (Kendall and Broughton, 1978).

In addition to WR5_B we analysed a 35 mm long drill core sample (WR_MC1) from an actively growing stalagmite that

was collected only a few meters away from WR5_B. WR_MC1 features a calcite fabric similar to that of WR5_B (Fig. A5). The bottom age of the drill core sample was dated at 3.45 ± 0.25 ka allowing us to reconstruct a late Holocene reference temperature for Whiterock cave that can be directly compared to temperatures derived from WR5_B.



## 2.2 Age model

U/Th dating of stalagmite WR5_B has previously been conducted by Meckler *et al.* (2012). The dating, however, was impaired by very low Uranium concentrations and several age reversals. Due to these difficulties, the initial age model of WR5_B was finally obtained by aligning the $\delta^{18}O_{cc}$ record of WR5_B with a refence record obtained from another, better dated Borneo stalagmite (GC08) that exhibits a very similar pattern of the $\delta^{18}O_{cc}$ signal. Stalagmite GC08 covers the time interval from 570 to 210 ka and originates from Green Cathedral Cave in Gunung Buda, located about 10 km northeast of

Whiterock Cave. Although GC08 contains significantly more Uranium, the initial $\delta^{234}U$ ratios were very low (–613 ‰ on average) and had to be assumed constant in order to derive a robust age model (Meckler et al., 2012). Resulting errors of the GC08 age model are therefore relatively large, on the order of ±12 kyr, for the time interval covered by WR5_B.

The $\delta^{18}O_{cc}$ records from both Borneo stalagmites display distinct maxima at glacial terminations T-V and T-IV that have been interpreted as deglacial drying phases (Meckler *et al.*, 2012). Similar maxima of the $\delta^{18}O_{cc}$ signal also occur in the

Asian Monsoon record (Cheng *et al.*, 2016) that is based on well dated stalagmites from Sanbao Cave (central China) with dating errors of about ±2.5 kyr at T-IV and ±1.5 kyr at T-V. Other Borneo stalagmites with better chronological constraints spanning the last glacial termination (T-I) show that the deglacial maximum in $\delta^{18}O_{cc}$ occurs simultaneously in the Chinese monsoon record and coincides with North Atlantic cooling during Heinrich stadial 1 (HS-1; Partin et al. 2007). Assuming the same synchronicity of the $\delta^{18}O_{cc}$ maxima in the monsoon and Borneo records during the older deglaciations covered by

WR5_B, we used the more accurate and precise age model of the Sanbao Cave stalagmites to improve the age model of WR5_B for the T-IV and T-V Heinrich stadials. The result is a shift of the initial WR5_B ages by about +4 kyr at T-IV and by –5 kyr at T-V (see Fig. A2). At Termination IV we additionally took account of a major hiatus (red dashed line in Fig. 2) that stands out due a clear discontinuity in the isotope $\delta^{18}O_{cc}$ record. Based on our match to the monsoon record, we estimate the duration of this growth interruption to about 3 kyr. We emphasise that these adjustments of the WR5_B age model were

restricted to the Heinrich stadials only, while ages during MIS12, MIS 11 and MIS 10 remain unaltered and thus are associated with relatively large age errors, including uncertainties arising from hiatuses of unknown duration. Moreover, a low average growth rate of 2.1 µm/yr limits the achievable temporal resolution of the WR5_B cave temperature record.

## 2.3 Fluid inclusion ages

To ascertain the ages of the fluid inclusions, the position of the analysed samples relative to the $\delta^{18}O_{cc}$ reference profile was

determined by visual correlation of prominent growth bands using high-resolution images of the thick sections and of the polished cut face of the stalagmite half on which the isotope transect had previously been drilled. While inclusion-rich layers appear brighter than inclusion-poor layers in reflected light on the (non-translucent) stalagmite half (Fig. 2), they appear darker in the translucent ca. 200 µm thick petrographic sections viewed in transmitted light (see Fig. A3, A5). To avoid potential confusion, we used an inverted grey-scale image of the stalagmite half (not shown) and compared it with the




conventional grey-scale images of the thick sections. Maximum errors of the visual layer correlation are estimated at ±1 mm. The ages of the individual samples were then determined based on the revised age model of WR5_B (Section 2.2). At each sample position, fluid inclusions were analysed within a narrow band of 1-3 mm width in axial (growth) direction of the stalagmite. Depending on the width of these bands and the stalagmite growth rate, the estimated maximum time intervals covered by a sample range between 0.2 and 3.0 kyr (see Table 1). However, since the age model and inferred growth rates do

not take account of unresolved growth interruptions some of these time intervals might be significantly shorter.

## 2.4 Fluid inclusion liquid-vapour homogenisation

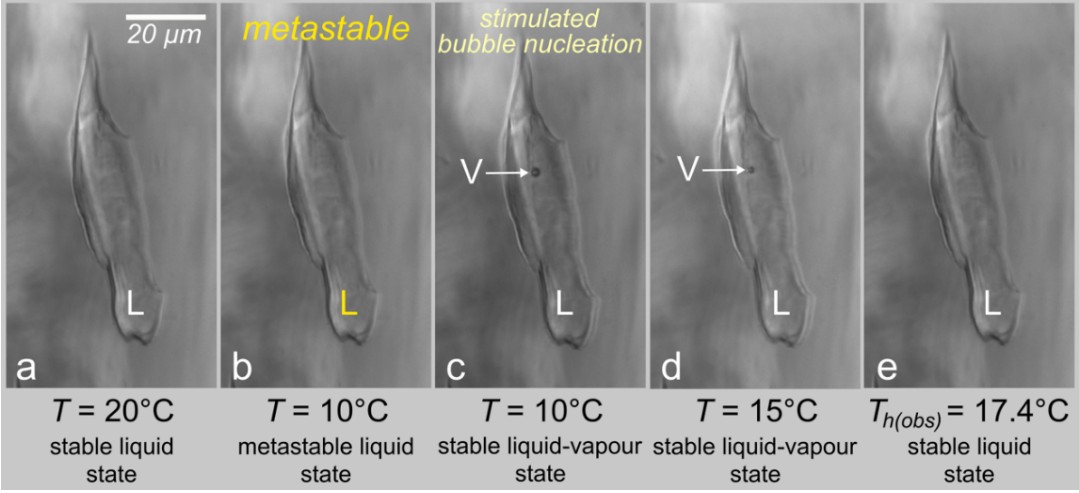

**Figure 4: Image sequence illustrating the different states (a-e) of a fluid inclusion during microthermometric measurements. After**
**transferring the inclusion from a stable into a metastable liquid state (from a to b), a single ultra-short laser pulse is applied to stimulate vapour bubble nucleation (c). Subsequently, the inclusion is heated, and the bubble becomes progressively smaller (d) until it collapses at $T_{h(obs)}$ (e), where the inclusion homogenizes into a stable liquid state. Modified after Krüger et al. (2014).**

Fluid inclusions in speleothems contain relicts of former drip waters from which calcite precipitated. The density $\varrho$ of the
enclosed water is assumed to relate directly to the cave temperature at the time the inclusions sealed off from the environment, and thus, can be used as a quantitative proxy to determine stalagmite formation temperatures $T_f$, *i.e.,* cave temperatures (Krüger *et al*., 2011; Løland et al., 2022). The density of the water in the inclusions can be deduced from measurements of the liquid-vapour homogenisation temperature $T_h$, using nucleation-assisted (NA) fluid inclusion microthermometry. Prior to measuring $T_h$, however, the initially monophase inclusions must be cooled below their expected
homogenisation temperature and then transferred from a metastable liquid to a stable liquid-vapour equilibrium state. To do so, we apply single ultra-short laser pulses to stimulate vapour bubble nucleation (Krüger *et al*., 2007), thus overcoming the long-lived metastability of the liquid state that prevents spontaneous bubble nucleation. Upon subsequent heating, the liquid phase expands at the expense of the vapour bubble that becomes continuously smaller. Eventually, the vapour bubble





collapses, and the inclusion homogenises to a stable liquid state. This temperature is referred to as observed homogenisation
temperature $T_{h(obs)}$. Fig. 4 displays an image sequence illustrating the different stages of a fluid inclusion during
microthermometric analysis.

## 2.5 Nucleation-assisted microthermometry

Microthermometric analyses were performed with a Linkam THMSG600 heating/cooling stage. The stage is mounted on an
Olympus BX53 microscope equipped with an Olympus LMPLFLN 100x/0.8 long working distance objective and a digital
camera (pco.edge 3.1) for visual observation of the inclusions in transmitted light. Synthetic $H_2O$ and $H_2O$-$CO_2$ fluid
inclusion standards are used for temperature calibration, yielding an accuracy of the heating/cooling stage of ±0.1 °C.
Reference temperatures are the triple point of water (0.0 °C) and the critical point of $CO_2$ (31.4 °C in the $H_2O$-$CO_2$ system;
Morrison, 1981).

For the microthermometric measurements we prepared separate, 200-300 μm thick unpolished sections that were removed
from the glass slides after cutting, and finally, split into fragments of about 5 mm in size that fit on the sample holder of the
Linkam stage. In order to make these samples transparent for microscopic observation of the fluid inclusions, they are coated
on both sides with a thin film of immersion oil to reduce light scattering at the raw cut faces.

Depending on the inclusion size, the samples were cooled 4 to 10 °C below the expected homogenisation temperature prior
to stimulating vapour bubble nucleation by means of ultra-short laser pulses provided by an amplified Ti:Sapphire
femtosecond laser system (CPA-2101, Clark-MXR, Inc.). The laser with a centre wavelength of 775 nm and a nominal pulse
width of < 150 fs is operated in single pulse mode, releasing a single laser pulse at the push of a trigger button. The pulse
energy is attenuated in three steps by approximately a factor of 50,000 before the laser beam is coupled into the microscope
via a short-pass dichroic beam splitter and focused on the fluid inclusion through the 100x objective. The last attenuation
step, consisting of a rotatable halfwave plate and a fixed Glan laser polarizer, allows for the fine-tuning of the pulse intensity
of the focused laser beam to a level that is at or slightly above the threshold for vapour bubble nucleation but below the
threshold for visible calcite ablation (Krüger *et al*., 2007). This is essential to avoid damaging of the inclusions, and thereby
altering their volume properties. A schematic representation of the analytical setup is shown in Fig. A3. The setup allows us
to repeatedly induce vapour bubble nucleation in pre-selected fluid inclusions under simultaneous visual observation and
to perform subsequent measurements of $T_{h(obs)}$ without moving the sample.

## 2.6 Analytical precision

At least two replicate measurements of $T_{h(obs)}$ were performed for each inclusion, yielding a typical reproducibility
(precision) within ±0.05 °C. Inclusions that did not homogenize below a certain threshold temperature (between 25-28 °C
depending on the climatic period) were rejected since they were likely subject to density alteration due to stretching or




partial leakage. Upper limits for $T_{h(obs)}$ measurements are used as a precaution to avoid potential stretching of other inclusions

in the sample caused by high internal fluid pressure that builds up when the inclusions are heated far above $T_{h(obs)}$.

As $T_{h(obs)}$ depends not only on the density of the enclosed water but also on the volume of the inclusions, we applied a correction to compensate for the effect of surface tension on liquid-vapour homogenisation (Fall et al., 2009). Based on $T_{h(obs)}$ and additional measurements of the vapour bubble radius $r(T)$ derived from bubble images taken at known temperature, we calculated the homogenisation temperature $T_{h\infty}$ of a hypothetical, infinitely large fluid inclusion using the

thermodynamic model proposed by Marti *et al.* (2012). The model is based on the IAPWS-95 formulation (Wagner and Pruß, 2002), and thus, in the strict sense, applies to a pure water system. The radius of the vapour bubble was calculated from circles that were graphically fitted to the bubble images using the *ImageJ* software. The average radius obtained from 3-4 bubble images was then used along with $T_{h(obs)}$ for calculating $T_{h\infty}$. Errors of the radius measurements translate to a temperature error that increases with decreasing inclusion volume and decreasing $T_{h\infty}$ (i.e., increasing water density). Based

on the spread of the measured bubble radii, the associated temperature error was typically found to be less than ±0.1 °C for the inclusions analysed in WR5_B. Another uncertainty relating to the calculation of $T_{h\infty}$ arises from the fact that $T_{h(obs)}$, i.e., the temperature at which the vapour bubble collapses, is thermodynamically not clearly defined. Marti *et al.* (2012) have demonstrated that the vapour bubble first becomes metastable before it ultimately becomes mechanically unstable. This means, the vapour bubble may collapse anywhere between the onset of the metastable two-phase state denoted as bubble

binodal ($T_{bin}$) and the stability limit of the two-phase state called bubble spinodal ($T_{sp}$). In the present study, we defined $T_{h(obs)}$ as the mean value between $T_{bin}$ and $T_{sp}$, which entails an additional uncertainty of $T_{h\infty}$ in the range of ±0.02 to ±0.14 °C depending on the volume and density of the inclusions. Altogether, for the inclusions analysed in WR5_B, the total uncertainty of $T_{h\infty}$ in terms of 2 standard deviation (2SD) is up to ±0.2 °C for inclusion volumes > $10^4$ μm³ and increases to about ±0.3 °C for inclusions of $10^3$ μm³. These values can be considered as analytical precision of $T_{h\infty}$.

**2.7 Cave temperature reconstruction**

In stalagmites, $T_{h\infty}$ can ideally be considered equal to the formation temperature $T_f$ of the inclusion and of the surrounding calcite host. However, the anticipated equality of $T_{h\infty}$ and $T_f$ builds on four fundamental assumptions underlying the application of fluid inclusion microthermometry to speleothems:

1. The density of the drip water trapped in fluid inclusions is controlled solely by cave temperature.
2. The fluid inclusions are closed systems that preserve the original physical and chemical properties of the enclosed drip water over geological times scales.
3. The closing age of the inclusions is equal to the age of the surrounding calcite.
4. The pure water system is an adequate approximation to describe the physico-chemical properties of natural cave drip waters.





If these assumptions were perfectly met, we would expect that fluid inclusions from the same stalagmite growth bands yield nearly identical $T_{h\infty}$ values within analytical error. And consequently, one single fluid inclusion would be sufficient to determine the cave temperature at a specific point in time. In reality, however, the scatter of $T_{h\infty}$ observed among apparently coeval fluid inclusions is much larger than the analytical precision (*cf.* Løland et al., 2022) and a larger number of inclusions needs to be analysed to assess the cave temperature on a statistical basis. In the present study we analysed in total 30 to 60

fluid inclusions per growth band ($n_{tot}$). The resulting $T_{h\infty}$ distributions are typically unimodal and Gaussian-like, and a 4 times Median Absolute Deviation criterion (4MAD; Hampel, 1974; Croux&Rousseeuw, 1992) was applied to exclude outliers from the statistical sample ($n_{stat} = n_{tot} -$ inclusions with $T_{h(obs)} >$ threshold). At the present stage of knowledge, we anticipate that the mean value of the $T_{h\infty}$ distribution of the outlier-adjusted net sample ($n_{net}$) represents the best approximation of the actual stalagmite formation temperature $T_f$ and cave temperature, respectively. Errors are reported as

standard error of the mean (2SEM) that relates to the precision of mean value and not to the accuracy of $T_f$.

## 3. Results

### 3.1 Microthermometry results from individual growth bands

In the present study we analysed a total of 2090 fluid inclusions in 49 different growth bands of stalagmite WR5_B (Fig. 2) and, in addition, 146 fluid inclusions in three growth bands of stalagmite WR_MC1. In both samples as well as in other

stalagmites from Northern Borneo (e.g. Løland *et al*., 2022), we observe a systematic scatter of $T_{h\infty}$ values of 4 - 6 °C within individual growth bands with average standard deviations (2SD) of 2 °C. This is about one order of magnitude larger than the analytical precision of the individual $T_{h\infty}$ values and significantly larger than natural cave temperature fluctuations that can be expected for the time intervals covered by a single growth band. The reasons for this variability of $T_{h\infty}$ in apparently coeval inclusions are subject of further investigations focusing on potential post-formation volume/density alterations of the

inclusions and variability of the closing ages due to temporarily persisting open porosity in the calcite fabric. Most of the analysed samples display unimodal, Gaussian-like distributions of the $T_{h\infty}$ values, suggesting that potential volume/density alterations affect $T_{h\infty}$ randomly and thus, are not supposed to impact the accuracy of the reconstructed cave temperatures. For two of the samples, however, the $T_{h\infty}$ distributions appear to be bimodal (Pos. 9 and 29). In these cases, the mean value of the dominant mode was considered representative for the stalagmite formation temperature, whereas the minor mode may result

from closing ages of some of the inclusions that were significantly younger than the age of the surrounding calcite host. Standard deviations (1SD) of the outlier-adjusted $T_{h\infty}$ distributions range from 0.43 to 1.61 °C, while standard errors of the mean (2SEM) range from 0.14 to 0.53 °C.





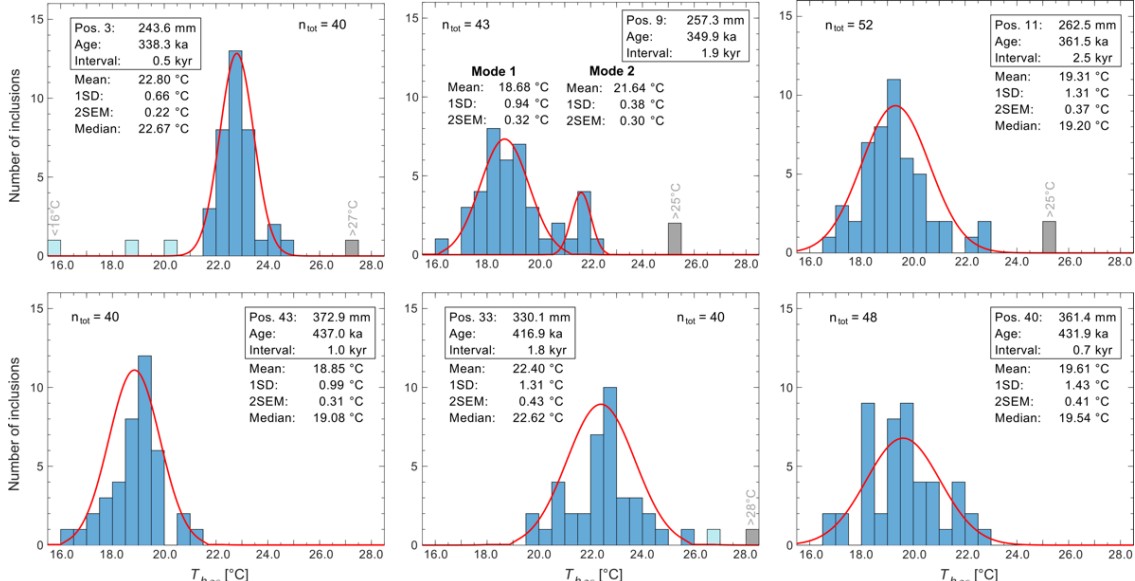

**Figure 4: Selection of $T_{h\infty}$ distributions obtained at different positions in stalagmite WR5_B. Inclusions that did not homogenize**
**below a pre-defined threshold temperature are shown in grey. Values labelled in light blue are considered outliers based on a**
**4MAD criterion and were excluded from further statistical analyses. Red curves are Gaussian distributions fitted to the data of the**
**remaining net sample ($n_{net}$, shown in blue) based on the mean and standard deviation (1SD). $n_{tot}$ is the total number of inclusions**
**that were analysed. The $T_{h\infty}$ distribution at sample position 9 was interpreted as bimodal and here the mean value of the dominant**
**lower mode is considered representative for the cave temperature at the time the surrounding calcite formed (see text).**

Reconstructed temperatures from WR5_B with statistical parameters are summarized in Table 1 along with the respective
sample positions, ages, and maximum time intervals covered by each sample. For an extended data summary, including $n_{tot}$,
$n_{net}$ and further statistical parameters, we refer to Table S1 in the supplementary material. A selection of $T_{h\infty}$ distributions at
different sample positions from WR5_B is shown in Fig. 4 and histogram plots from all 49 sample positions, together with
additional plots of the cumulative distribution and $T_{h\infty}$-volume diagrams are provided in Fig. A4. The respective numerical
data are reported in Table S2 (supplementary material). The WR_MC1 drill core sample, on the other hand, yields an
average late Holocene temperature for Whiterock Cave of 23.28 ±0.25 °C (2SEM), slightly lower than the present-day cave
temperature of 23.7 ±0.1 °C. The results derived from three growth bands are illustrated in Fig. A5 and numerical values are
reported in Table S3 (supplementary material).








**Table 1: Summary of results from WR5_B.**

The table lists the mean position (in mm) of the analysed growth bands relative to the isotope transect, the mean sample age and the maximum time interval covered by the sample. The statistical sample ($n_{stat}$) is the number of inclusions that homogenised below a given threshold value. Data that were rejected from the statistical sample based on a 4MAD outlier criterion are indicated in percentage. Cave temperature $T_{cave}$ is defined as the average value of the respective $T_{h\infty}$ distribution and reported with 1SD and 2SEM. Cave temperatures were finally corrected for the secondary effect of sea level-related cave altitude changes using the reconstruction of Bintanja et al. (2005) and a lapse rate of 0.6 °C/100 m (Løland et al., 2022), yielding $T_{corr}$ (see Section 3.2).

| Sample name | Sample Position no. | Mean Position (mm) | Mean Age (ka) | Time Interval (kyr) | statistical sample¹ $n_{stat}$ | rejected data² % | $T_{h\infty\_avg}$ ≡ $T_{cave}$ (°C) | 1 SD (°C) | 2 SEM (°C) | Sea Level³ (mean) (m) | 1 SD (m) | $T_{corr.}$ s.l. corr (°C) | total error (°C) |
|---|---|---|---|---|---|---|---|---|---|---|---|---|---|
| WR5_B1_2 | 1 | 233.8 | 334.15 | 0.86 | 44 | 0.0 | 23.64 | 1.01 | 0.31 | -30.29 | 6.82 | 23.83 | 0.31 |
| WR5_B1_3 | 2 | 238.9 | 336.33 | 0.98 | 33 | 9.1 | 23.51 | 0.82 | 0.30 | -60.55 | 8.43 | 23.88 | 0.31 |
| WR5_B1_1 | 3 | 243.6 | 337.88 | 0.20 | 39 | 7.7 | 22.80 | 0.66 | 0.22 | -79.78 | 9.60 | 23.28 | 0.24 |
| WR5_B1_4 | 4 | 248.5 | 338.58 | 1.00 | 79 | 0.0 | 22.17 | 1.03 | 0.23 | -86.57 | 10.01 | 22.68 | 0.25 |
| WR5_B1_8 | 5 | 250.5 | 340.16 | 0.43 | 58 | 0.0 | 22.44 | 1.04 | 0.27 | -104.05 | 9.63 | 23.06 | 0.30 |
| WR5_B1_5 | 6 | 252.8 | 343.67 | 2.00 | 67 | 3.0 | 19.78 | 0.95 | 0.23 | -110.31 | 8.72 | 20.44 | 0.26 |
| WR5_B1_6 | 7 | 254.2 | 345.60 | 0.69 | 43 | 4.7 | 19.63 | 0.78 | 0.24 | -105.75 | 8.63 | 20.27 | 0.27 |
| WR5_B1_7 | 8 | 255.3 | 347.12 | 1.38 | 44 | 2.3 | 18.88 | 0.68 | 0.21 | -102.39 | 8.67 | 19.49 | 0.24 |
| WR5_B1_9 | 9 | 257.3 | 349.89 | 1.94 | 41 | 0.0 | 19.14 | 1.42 | 0.44 | | | | |
| *mode 1* | | *257.3* | *349.89* | | | | *18.68* | *0.94* | *0.32* | *-96.80* | *8.90* | *19.26* | *0.34* |
| *mode 2* | | *257.3* | *349.89* | | | | *21.64* | *0.38* | *0.30* | | | | |
| WR5_B1_10 | 10 | 260.2 | 356.32 | 2.49 | 40 | 0.0 | 18.99 | 0.81 | 0.26 | -88.61 | 8.33 | 19.52 | 0.28 |
| WR5_B1_11 | 11 | 262.5 | 361.45 | 2.47 | 50 | 0.0 | 19.31 | 1.31 | 0.37 | -81.75 | 8.74 | 19.80 | 0.38 |
| WR5_B2 (a) | 12 | 267.4 | 363.67 | 0.32 | 25 | 4.0 | 20.37 | 0.86 | 0.35 | -75.74 | 8.58 | 20.82 | 0.36 |
| WR5_B2 (b) | 13 | 268.2 | 364.03 | 0.58 | 49 | 0.0 | 20.20 | 0.73 | 0.21 | -74.70 | 8.56 | 20.65 | 0.23 |
| WR5_B2 (c) | 14 | 270.4 | 365.86 | 2.09 | 50 | 0.0 | 19.88 | 0.91 | 0.26 | -70.77 | 8.35 | 20.30 | 0.27 |
| WR5_B2 (d) | 15 | 273.5 | 368.59 | 1.85 | 58 | 1.7 | 19.67 | 0.92 | 0.24 | | 8.09 | 20.07 | 0.26 |
| WR5_B2 (e) | 16 | 279.2 | 373.60 | 2.02 | 54 | 0.0 | 19.83 | 1.02 | 0.28 | -64.13 | 8.20 | 20.21 | 0.29 |
| WR5_B2 (f) | 17 | 283.4 | 377.30 | 1.41 | 26 | 7.7 | 19.40 | 0.84 | 0.34 | -62.04 | 9.12 | 19.77 | 0.35 |
| WR5_B2 (g) | 18 | 285.2 | 378.88 | 1.32 | 44 | 0.0 | 20.71 | 1.31 | 0.39 | -61.54 | 9.46 | 21.07 | 0.40 |
| WR5_B2 (h) | 19 | 287.4 | 380.82 | 0.97 | 31 | 3.2 | 20.12 | 0.90 | 0.33 | -60.26 | 9.59 | 20.48 | 0.34 |
| WR5_B3_1 | 20 | 289.9 | 383.02 | 2.37 | 43 | 0.0 | 20.13 | 1.12 | 0.34 | -57.66 | 8.45 | 20.48 | 0.35 |
| WR5_B3_8 | 21 | 293.3 | 386.67 | 1.29 | 40 | 2.5 | 21.52 | 1.17 | 0.37 | -49.44 | 5.72 | 21.82 | 0.38 |
| WR5_B3_4 | 22 | 295.0 | 388.65 | 0.93 | 37 | 0.0 | 21.64 | 1.06 | 0.35 | -44.73 | 4.48 | 21.91 | 0.35 |
| WR5_B3_9a | 23 | 295.9 | 389.70 | 1.40 | 45 | 2.2 | 21.59 | 1.26 | 0.38 | -38.39 | 2.63 | 21.82 | 0.38 |
| WR5_B3_9b | 24 | 297.0 | 390.98 | 1.28 | 37 | 2.7 | 21.64 | 1.01 | 0.34 | -33.97 | 1.84 | 21.85 | 0.34 |
| WR5_B3_5 | 25 | 298.7 | 392.96 | 1.63 | 36 | 0.0 | 22.35 | 0.71 | 0.24 | -27.38 | 1.15 | 22.42 | 0.24 |
| WR5_B3_10 | 26 | 302.0 | 396.80 | 2.56 | 42 | 0.0 | 22.84 | 0.91 | 0.28 | -6.36 | 0.30 | 22.88 | 0.28 |
| WR5_B3_6 | 27 | 307.2 | 400.58 | 1.43 | 39 | 7.7 | 23.28 | 0.43 | 0.14 | -0.14 | 1.25 | 23.28 | 0.14 |
| WR5_B3_2 | 28 | 311.9 | 403.94 | 1.43 | 30 | 3.3 | 24.09 | 1.16 | 0.43 | 0.28 | 5.78 | 24.09 | 0.43 |
| WR5_B3_12 | 29 | 314.2 | 404.87 | 1.00 | 34 | 2.9 | 23.46 | 1.36 | 0.47 | | | | |
| *mode 1* | | *314.2* | *404.87* | | | | *21.73* | *0.75* | *0.46* | | | | |
| *mode 2* | | *314.2* | *404.87* | | | | *24.21* | *0.62* | *0.26* | *0.21* | *6.43* | *24.21* | *0.26* |
| WR5_B3_11 | 30 | 317.5 | 407.23 | 1.86 | 39 | 0.0 | 23.33 | 0.84 | 0.27 | 0.11 | 8.25 | 23.33 | 0.27 |
| WR5_B3_3 | 31 | 321.2 | 410.59 | 1.50 | 37 | 2.7 | 23.83 | 0.67 | 0.22 | -3.65 | 9.82 | 23.85 | 0.23 |
| WR5_B3_7 | 32 | 324.1 | 412.66 | 1.93 | 36 | 5.6 | 23.54 | 0.68 | 0.23 | -12.91 | 8.21 | 23.62 | 0.24 |
| WR5_B4_1 | 33 | 330.1 | 416.95 | 1.79 | 39 | 2.6 | 22.40 | 1.31 | 0.43 | -33.03 | 9.03 | 22.60 | 0.43 |
| WR5_B4_2 | 34 | 334.2 | 419.88 | 2.43 | 40 | 0.0 | 21.91 | 1.11 | 0.35 | -44.95 | 8.76 | 22.18 | 0.36 |
| WR5_B4_5 | 35 | 342.2 | 425.54 | 3.00 | 37 | 0.0 | 22.28 | 1.61 | 0.53 | -86.29 | 9.72 | 22.80 | 0.54 |
| WR5_B4_3 | 36 | 346.7 | 427.00 | 0.47 | 36 | 2.8 | 21.40 | 0.92 | 0.31 | -100.98 | 10.53 | 22.00 | 0.33 |
| WR5_B4_6 | 37 | 349.7 | 427.88 | 0.56 | 38 | 2.6 | 20.92 | 1.36 | 0.45 | -107.32 | 10.88 | 21.56 | 0.46 |
| WR5_B4_4 | 38 | 355.0 | 429.44 | 0.62 | 38 | 2.6 | 20.01 | 1.07 | 0.35 | -120.31 | 10.85 | 20.73 | 0.38 |
| WR5_B4_8 | 39 | 357.2 | 430.11 | 0.41 | 41 | 0.0 | 19.82 | 1.34 | 0.42 | -123.64 | 10.88 | 20.56 | 0.44 |
| WR5_B5_7 | 40 | 361.4 | 431.90 | 0.67 | 48 | 0.0 | 19.61 | 1.43 | 0.41 | -126.94 | 10.59 | 20.37 | 0.44 |
| WR5_B5_1 | 41 | 364.7 | 433.36 | 1.07 | 40 | 0.0 | 18.74 | 0.89 | 0.28 | -126.78 | 10.57 | 19.50 | 0.31 |
| WR5_B5_2 | 42 | 369.3 | 435.41 | 1.07 | 41 | 0.0 | 18.49 | 1.07 | 0.34 | -123.70 | 10.19 | 19.24 | 0.36 |
| WR5_B5_3 | 43 | 372.9 | 437.01 | 0.98 | 40 | 0.0 | 18.85 | 0.99 | 0.31 | -118.73 | 9.54 | 19.56 | 0.34 |
| WR5_B5_4 | 44 | 383.6 | 441.76 | 1.64 | 41 | 4.9 | 19.07 | 0.75 | 0.24 | -111.47 | 9.76 | 19.74 | 0.27 |
| WR5_B5_6 | 45 | 387.2 | 443.36 | 1.20 | 38 | 2.6 | 19.19 | 0.93 | 0.31 | -109.46 | 9.71 | 19.85 | 0.33 |
| WR5_B5_5 | 46 | 391.8 | 445.41 | 0.84 | 44 | 0.0 | 19.02 | 1.14 | 0.34 | -104.24 | 10.06 | 19.64 | 0.37 |
| WR5_B6_8_1 | 47 | 406.3 | 451.85 | 1.16 | 36 | 2.8 | 19.36 | 0.68 | 0.23 | -94.57 | 10.79 | 19.93 | 0.26 |
| WR5_B6_8_2 | 48 | 414.6 | 455.54 | 1.73 | 27 | 0.0 | 19.51 | 1.35 | 0.52 | -91.53 | 10.00 | 20.06 | 0.53 |
| WR5_B6_8_3 | 49 | 423.0 | 459.27 | 1.19 | 40 | 0.0 | 19.51 | 1.32 | 0.42 | -90.52 | 8.15 | 20.06 | 0.43 |
| | | | TOTAL | | 2034 | | | | | | | | |
| | | | Average | | 41.5 | 1.9 | | 1.00 | 0.32 | | | | 0.33 |
| | | | MIN | | | | | 0.38 | 0.14 | | | | 0.14 |
| | | | MAX | | | | | 1.61 | 0.53 | | | | 0.54 |

¹ statistical sample = total number of inclusions − number of inclusions with Th(obs) > threshold value
² rejected from the statistical sample based on a 4MAD outlier criterion (Median Absolute Deviation)
³ based on the sea level reconstruction of Bintanja et al. (2005)

## 3.2 Borneo temperature record

The reconstructed Whiterock Cave temperature record is illustrated in Fig. 5b and numerical values are reported in Table 1. In order to assess the amplitude of the climate-driven glacial-interglacial temperature changes – one of the objectives of this study – the cave temperatures were corrected to eliminate the secondary temperature effect induced by changes of the cave altitude relative to sea level that was about 120 m lower during glacial maxima. To do so, we used the global sea level reconstruction of Bintanja *et al.* (2005) illustrated in Fig. 5a and assumed a constant lapse rate of surface air temperatures of 0.6 ± 0.1°C/100 m (Løland *et al.*, 2022). The correction is largest at the glacial maxima and raises the reconstructed cave temperature by up to 0.8 °C. The temperature record shown in Fig. 5b displays both the actual cave temperatures $T_{cave}$ (blue)




and sea level corrected temperatures $T_{corr.}$ (green). Propagated errors of $T_{corr.}$ are up to 14 % larger than 2SEM of $T_{cave}$ at
glacial maxima and include the assigned uncertainty of the lapse rate that possibly varies with climatic changes (Loomis *et al*. 2017), as well as uncertainties of the employed sea level reconstruction (see Table 1). A comparison of global sea level reconstructions of Bintanja *et al.* (2005) and Spratt & Lisiecki (2016) yielded only minor differences in the resulting temperature corrections (Fig. A6, Table B1).

The temperature record shows a clear pattern of glacial-interglacial temperature changes. Average $T_{corr.}$ values during glacial
maxima of MIS 12 and MIS 10 were at 19.6±0.3 °C (n = 6) and 19.5±0.3 °C (n = 4), respectively, while average temperatures at the interglacial optimum of MIS 11 were 23.8±0.3 °C (n = 5), which results in an amplitude of glacial-interglacial land temperature changes of 4.2±0.4 °C. Considering the extreme values only, 19.2±0.4 °C (MIS 12), 24.2±0.3 °C (MIS 11), and 19.3±0.3 °C (MIS 10), the maximum amplitude would be 5.0±0.5 °C.

The WR5_B record displays a well-resolved continuous temperature increase across the initial stage of T-V reaching a
preliminary maximum at 22.8±0.5 °C around 425 ka, which corresponds to a warming of 3.2±0.6 °C. This initial temperature increase occurs concurrent with a strong Heinrich event (T-V HS) in the Northern Hemisphere (Vázques Riveiros *et al.* 2013) and a phase of deglacial drying in Northern Borneo as manifested in the stalagmite $\delta^{18}O_{cc}$ signal (Fig. 5c). Since $\delta^{18}O_{cc}$ and temperature were derived from the same stalagmite, we emphasize that this temporal correlation is independent of any age uncertainties. After reaching a maximum at the end of the Heinrich stadial, the temperature decreases slightly, passes a
minimum, similar to the Antarctic Cold Reversal (ACR) during Termination I, and subsequently further increases towards the MIS 11 interglacial optimum. For Termination T-IV, in contrast, the recorded evolution of deglacial warming is incomplete due to a hiatus of about 3 kyr (see Section 2.2 and Fig. A2). This hiatus manifests itself not only petrographically (Fig. A1a and h) but also through a clear discontinuity both in the temperature and $\delta^{18}O_{cc}$ records (Fig. 5b and c). Substantial growth interruptions and/or calcite dissolution might also be the reason for very low average growth rates (0.4 – 0.7 µm/yr)
in the run-up to T-IV. The WR5_B record ends with a deglacial temperature maximum of 23.8±0.3 °C (n = 2) at the end of the Heinrich stadial that coincides with the early MIS 9 interglacial optimum. Again, deglacial warming occurs during the T-IV HS, simultaneous with a pronounced excursion of the $\delta^{18}O_{cc}$ signal towards more positive values, indicating drier climate. The amplitude of deglacial warming across the T-IV HS is 4.3±0.4 °C, which is about 1.1 °C higher than the temperature increase across the corresponding Heinrich stadial at T-V (3.2±0.6 °C).

**3.3 Hydroclimate**

The calcite oxygen isotope record ($\delta^{18}O_{cc}$) of stalagmite WR5_B has previously been interpreted to reflect primarily changes in hydroclimate and precipitation amount (Meckler et al. 2012) assuming that the effect of temperature on the $\delta^{18}O_{cc}$ signal is relatively minor. As a side-product of this study, we employed our temperature record to calculate the oxygen isotopic composition of the calcite supplying drip water $\delta^{18}O_{dw}$ from which the stalagmite formed. For the calculations we used the



uncorrected cave temperatures $T_{cave}$ and $\delta^{18}O_{cc}$ together with the empirical speleothem calibration for the calcite-water

oxygen isotope fractionation proposed by Tremaine et al. (2011). The calculated $\delta^{18}O_{dw}$ record (Fig. 5d) agrees overall well

with measured $\delta^{18}O_{FI}$ values from previous fluid inclusion isotope analysis (Meckler et al. 2015), with few notable

deviations at T-V and during part of MIS11.

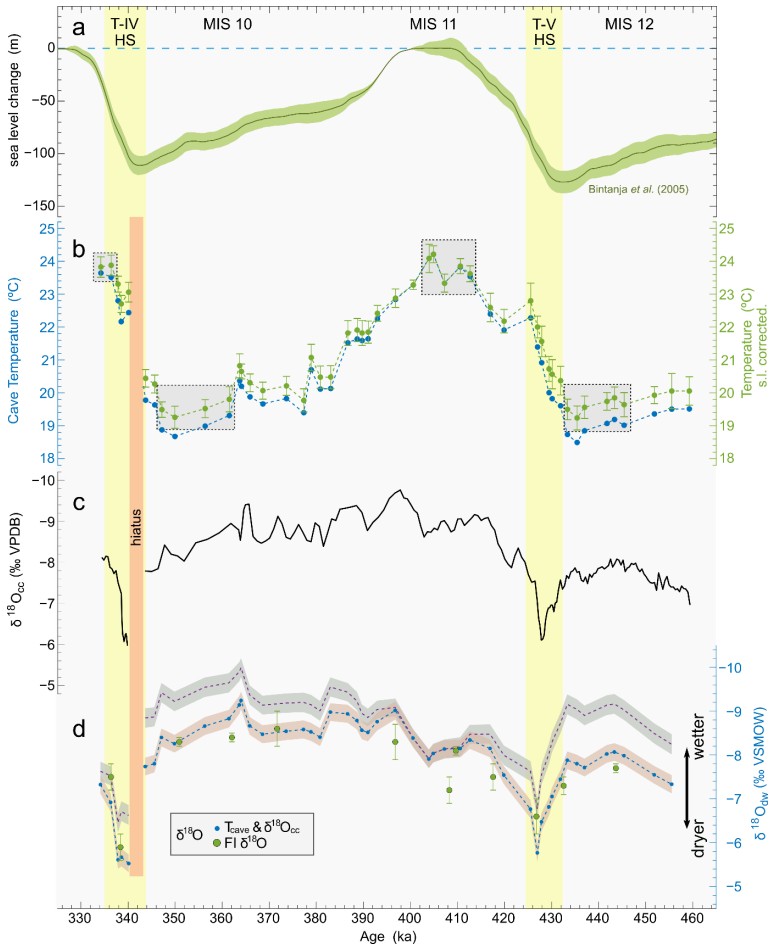


**Fig. 5: a) Sea level reconstruction with 1SD (Bintanja *et al.*, 2006), plotted on the LR04 time scale (Lisiecki & Raymo, 2005). Note the lag of sea level rise relative to temperature at the terminations. Yellow bars indicate T-IV and T-V Heinrich stadials. b)**
**WR5_B temperature record (this study). Blue dots represent actual cave temperatures ($T_{cave}$) and green dots are the respective sea level corrected temperatures ($T_{corr.}$). For graphical reasons, error bars (2SEM) are indicated for $T_{corr.}$ only. Grey boxes indicate the $T_{corr}$ data that were used for calculating glacial and interglacial mean temperatures. c) WR5_B oxygen isotope record ($\delta^{18}O_{cc}$) of the stalagmite calcite (Meckler *et al.*, 2012), d) reconstructed drip water isotope record $\delta^{18}O_{dw}$ (small blue dots) derived from $T_{cave}$ and $\delta^{18}O_{cc}$ using the empirical calibration for oxygen isotope fractionation in speleothems of Tremaine et al. (2011) plotted with 95% CI (light brown band). Green dots are measured $\delta^{18}O_{FI}$ values from fluid inclusion water analysis (Meckler *et al.*, 2015, data**
**from Bern lab) with error bars indicating 1SD of 2-3 replicate measurements. The purple dashed line with the grey 95% CI band shows the ice volume correction that shifts glacial $\delta^{18}O_{dw}$ to more negative values. Note, in both c) and d) $\delta^{18}O$ is plotted on reverse axes.**



The calculated $\delta^{18}O_{dw}$ record was then corrected for global ice volume changes to account for changes of the sea water oxygen isotopic composition using again the sea level reconstruction of Bintanja *et al.* (2012) and a correction factor of 1

‰/100 m sea level change (Schrag et al., 2002). As a result, glacial $\delta^{18}O_{dw}$ shifts to more negative values (purple curve in Fig. 5d). The calculated record confirms the previous interpretation of relative drying during the T-V and T-IV Heinrich stadials inferred from $\delta^{18}O_{cc}$. Moreover, in consequence of the ice volume correction, a glacial-interglacial difference in $\delta^{18}O_{dw}$ arises, indicating relatively wetter conditions during the glacial periods (MIS 10 and 12) compared to the interglacial (MIS 11). We note, however, that the correction for global ice volume effect does not take into account potential additional

variations in sea water $\delta^{18}O$ in the moisture source regions.

## 4. Discussion

### 4.1 Comparison with Antarctic ice core records

Antarctic ice cores provide a wealth of different proxies for reconstructing past climate back to 800 ka. For comparison with our Borneo temperature record, we focus on two EPICA (European Project for Ice Coring in Antarctica) Dome C (EDC) ice

core records, namely a recently published record of atmospheric $CO_2$ concentrations (Nehrbass-Ahles *et al.*, 2020), and two temperature records: the classical EDC record of surface temperature anomalies ($\Delta T_s$) derived from hydrogen isotope measurement of the ice, $\delta D_{ice}$ (Jouzel *et al.*,2007), and a more recent $\Delta T_{site}$ reconstruction based on d-excess, i.e., using both $\delta D_{ice}$ and $\delta^{18}O_{ice}$ (Landais et al., 2021). These EDC records (Fig. 6a,b) were plotted on the AICC2012 time scale (Bazin *et al.*, 2013) and demonstrate the close correlation between Antarctic temperature anomalies and atmospheric $CO_2$

concentrations. The $CO_2$ record of Nehrbass-Ahles *et al.* (2020) comprises a 148 kyr time interval from 297 to 445 ka, and hence, does not cover the older MIS12 period included in the WR5_B record. To fill this 15 kyr gap we used previous data from Siegenthaler et al. (2005) that were also employed for the 800 kyr $CO_2$ composite record of Bereiter *et al.* (2015). The two EDC temperature records differ systematically during glacial periods with slightly higher temperatures (smaller $\Delta T$) in the reconstruction of Landais et al. (2021), whereas interglacial temperature anomalies are nearly identical.

Despite some age uncertainties of our Borneo temperature record, we found generally good agreement of the tropical land temperature record with the structure of the Antarctic $\Delta T$ records. For the T-V HS, for which we can assume age uncertainties comparable to those of the AICC2012 ice core chronology, the timing of deglacial warming is nearly identical at both locations, exhibiting a maximum at the end of the Heinrich stadial, followed by a slight decrease of temperature and atmospheric $CO_2$. The amplitude of Antarctic temperature changes, however, is about a factor 2.3 larger compared to Borneo

land temperature changes (see Section 4.4). The ice core records also display a series of millennial scale $CO_2$ and temperature fluctuations during the glacial inception and MIS 10. Similar patterns are discernible in the Borneo record; however, a clear correlation of the individual events is hampered by substantial age uncertainties in this part of the record





and potential unconstrained hiatuses in the stalagmite record. For the T-IV HS, the WR5_B record provides reliable information only about the amplitude of the deglacial warming (4.3 ±0.4 °C) as only the high temperature end of the deglacial warming is recorded.

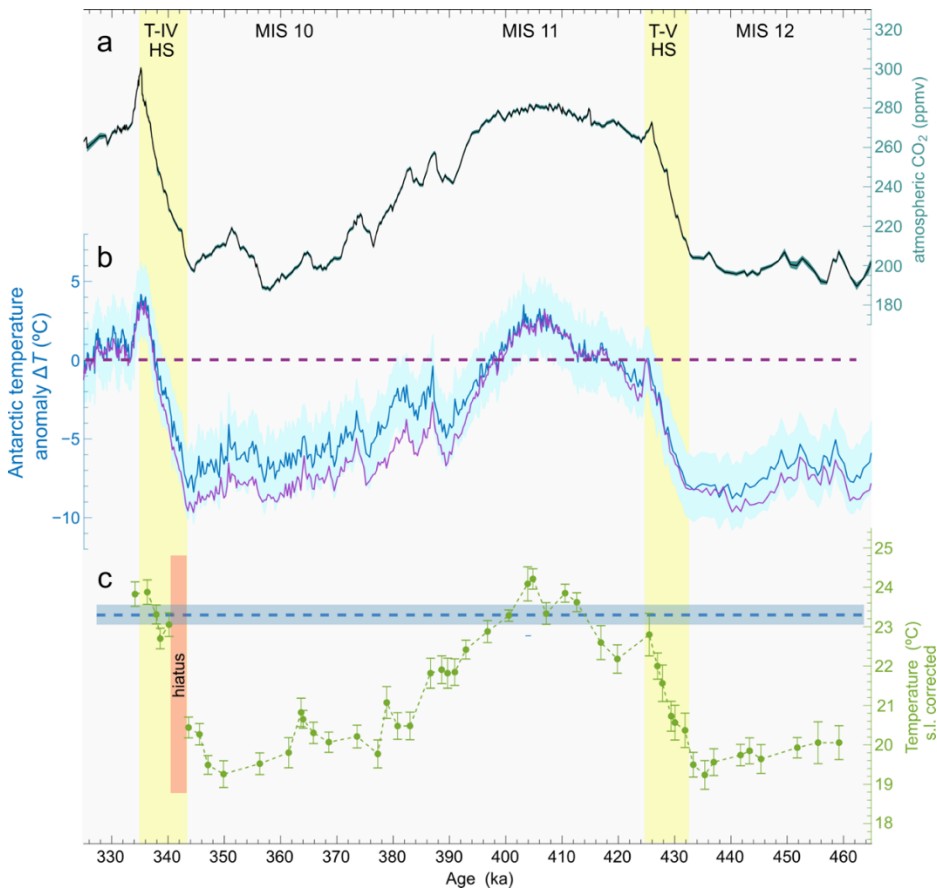

**Fig. 6: Comparison of the Borneo temperature record (*T*$_{corr.}$) with Antarctic ice core (EDC) records: a) atmospheric CO$_2$ concentrations (Nehrbass-Ahles *et al.*, 2020 and Siegenthaler et al., 2005). b) Antarctic temperature anomalies *ΔT* (purple: Jouzel et al., 2007; blue: Landais et al., 2021 with ± 2 °C uncertainty). The dashed purple line indicates *ΔT* = 0 °C. c) Borneo temperature record (*T*$_{corr.}$) derived from stalagmite WR5_B corrected for sea level-related altitude changes. The dashed blue line indicates the late Holocene reference temperature in Whiterock cave (23.3 ± 0.25 °C) determined from stalagmite WR_MC1 (Fig. S4)). All EDC records are plotted on the AICC2012 ice core chronology. Yellow bars indicate the T-V and T-IV Heinrich stadials.**

Peak temperatures in the EDC *ΔT* records for the interglacial optimum (MIS 11) and for MIS 9 (occurring at the end of the T-IV HS) were 2.5 to 3.5 °C higher than the Holocene reference value (*ΔT* = 0, dashed horizontal line in Fig. 6b). For the Borneo record, we compared the respective peak temperatures (24.2 ±0.3 °C and 23.9 ±0.3 °C) with the late Holocene reference temperature of 23.28 ±0.25 °C (dashed horizontal line in Fig. 6c) derived from the drill core sample WR_MC1. We thus infer that the temperature maximum of MIS 11 was about 0.9 ± 0.4 °C warmer compared to the late Holocene,





which is roughly consistent with the previously mentioned polar amplification factor of 2.3. The peak temperature at the end
of T-IV, in contrast, is only 0.6 ± 0.4 °C warmer than late Holocene, compared to 3.5 °C in the Antarctic record, thus
suggesting either a much stronger temperature overshoot in Antarctica compared to tropical Borneo or, more likely, that we
missed the MIS 9 peak temperature in our record.

### 4.2 Comparison with sea surface temperatures (SSTs)

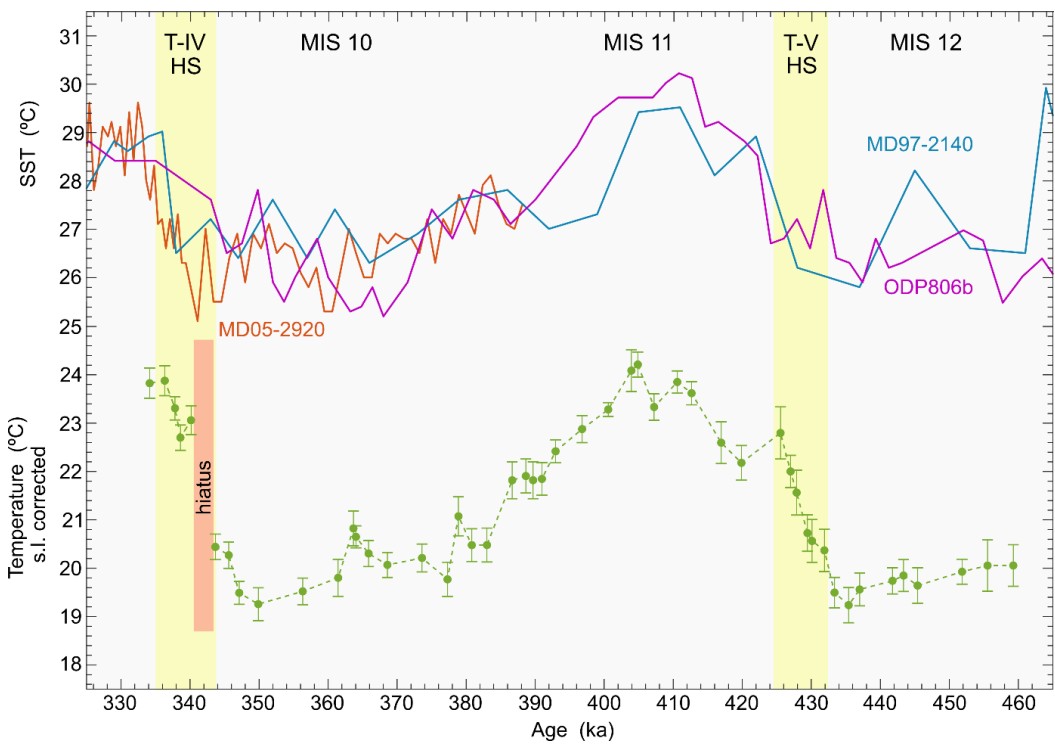

**Fig. 7: Comparison of the WR5_B land temperature record ($T_{corr.}$) with Mg/Ca based sea surface temperatures (SSTs) from three
different sites in the Western Pacific warm pool (see Fig. 1 for locations).**

We compare our Borneo land temperature record with Mg/Ca based sea surface temperature (SST) records derived from
three Western Pacific core sites (Fig. 7), namely ODP Site 806 (Medina-Elizalde&Lea, 2005), MD97-2140 (de Garidel-
Thoron et al., 2005), and MD05-2920 (Tachikawa et al., 2014). The locations of the core sites are shown in Fig 1. The first
two records were reported on an orbitally tuned reference chronology established on ODP Site 677 (Shackleton et al., 1990),
while the latter is reported on the LR04 timescale (Lisiecki&Raymo, 2005). The two SST records from Site 806 and MD97-
2140 have relatively low temporal resolution but, in contrast to the higher resolved MD05-2920 record, cover the entire time
interval of the WR5_B record. Although the SST data appear more variable compared to cave temperatures, partly owing to
the lower temporal resolution, they clearly show the general glacial-interglacial temperature trend with differences of about 4
°C between MIS 12 and MIS 11, whereas deglacial warming across T-IV is less than 3 °C in the ODP806 and MD97-2140





records. At site MD05-2920, in contrast, the amplitude of temperature change across T-IV reaches about 4 °C, which is consistent with the amplitude obtained in the WR5_B record. In terms of absolute temperatures, it is worth noting that present-day SSTs around Northern Borneo are about 5 °C higher than cave temperatures (Meckler et al., 2015) and a similar
offset is seen in the records for MIS 12-9.

## 4.3 Comparison of T-V and T-I

Recently, Løland et al. (2022) published a cave temperature record from Northern Borneo based on fluid inclusion microthermometry covering the last glacial Termination (T-I). Stalagmite SC02 that was used for the study was collected from Secret Chamber (ca. 105 m a.s.l.), in the Clearwater cave system, which is about 8 km from Whiterock Cave. Their
reconstructed temperature record ($T_{corr.}$) is shown in Fig. 8 together with calcite $\delta^{18}O_{cc}$ (Buckingham et al. 2022) and the corresponding record of Antarctic temperature anomalies $\Delta T_s$ (Jouzel et al., 2007) for comparison. When compared to our new temperature reconstruction across T-V, the two records exhibit the same structure and timing. Both show an early deglacial warming starting at the beginning of Heinrich stadials, a preliminary maximum at the end of the Heinrich events, followed by a slight temperature decrease and a subsequent recovery. At T-I, the excursion of $\delta^{18}O_{cc}$ to more positive values,
indicating relatively drier conditions, is clearly related to the H-1 interval and the analogous behaviour is observed during the Heinrich stadial at T-V. The similar pattern of early deglacial warming concurrent with relative drying in Borneo during deglacial Heinrich events, when the Northern Hemisphere cools, corroborates the finding of Løland et al. (2022) that during deglaciations, the temperature evolution in Northern Borneo proceeds in step with Southern Hemisphere temperature and atmospheric $CO_2$ concentrations while hydroclimate is affected by Northern Hemisphere cooling.
Early deglacial warming in Antarctica has been attributed to the classical seesaw behaviour, with heat release or accumulation in the Southern Ocean caused by a weakening of the Atlantic Meridional Overturning Circulation (AMOC) associated with reduced heat transport to and deep ocean ventilation in the North (e.g., Broecker, 1998; Pedro et al., 2011). The subsequent slight cooling (e.g., during the ACR) then corresponds to the re-establishment of the AMOC. The fact that the same timing of deglacial warming can also be observed in tropical Borneo demonstrates that the temperature response to
a reduced AMOC is not restricted to the high southern latitudes and Antarctica.

The amplitude of deglacial warming in Northern Borneo during H-1 was 2.6 ±0.3 °C, compared to 3.2 ±0.6 °C during the corresponding Heinrich stadial at T-V, while the total temperature increase from the Last Glacial Maximum (LGM) to the early-to-mid Holocene amounts to 3.6 ±0.3 °C (Løland et al., 2022), compared to 4.2 ± 0.4 °C from MIS 12 to the MIS 11 temperature optimum. When correcting SC02 temperatures for the difference in cave altitude between Secret Chamber and
Whiterock Cave by 0.3±0.14 °C based on the offset of present-day cave temperatures (24.0 ± 0.1 °C and 23.7 ± 0.1 °C, respectively), the altitude-corrected early-to mid-Holocene average temperature from SC02 would be 23.1 ± 0.28 °C, and thus 1.1 ± 0.4 °C colder than the MIS 11 temperature optimum derived from WR5_B. This is in line with the temperature difference of 0.9 ± 0.4 °C between MIS11 and the late Holocene reference temperature (23.3 ± 0.25 °C) determined for



Whiterock cave (see section 4.1) and demonstrates the consistency of the reconstructed temperature data across different
stalagmites and caves. Using the same 0.3 centigrade altitude correction, the average LGM temperature derived from SC02
decreases to 19.5 ± 0.35 °C (n=3), equivalent to the average temperatures at the glacial maxima of MIS 12 (19.6±0.3 °C; n =
6) and MIS 10 (19.5±0.3 °C; n = 4). While our new record reveals warmer interglacial temperatures during MIS11 compared
to the Holocene, there is no significant temperature difference between the three glacial maxima, which is in line with
Antarctic $\Delta T$ reconstructions.

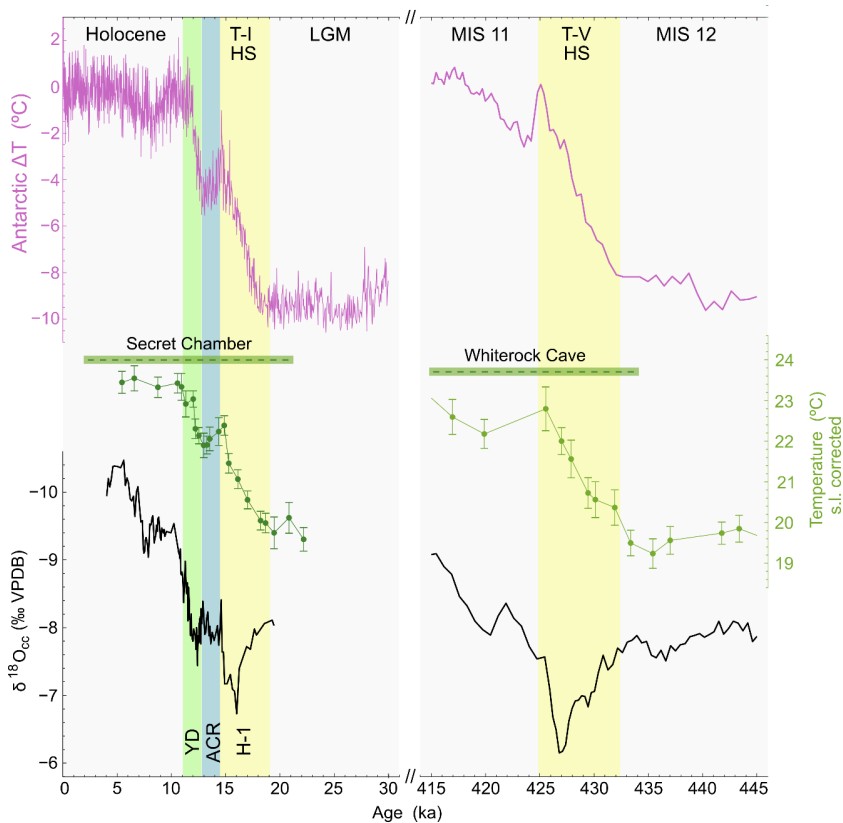


**Fig. 8: Comparison of T-I and T-V. Top: Antarctic temperature anomalies $\Delta T_s$ (Jouzel et al., 2007) plotted on the AICC2012 ice core chronology (Veres et al., 2013 and Bazin et al., 2013). Middle: SC02 temperature reconstruction ($T_{corr}$) across T-I (Løland et al. 2022; not corrected for differences in cave altitude) and WR5_B record across T-V (this study). Error bars indicate 2SEM.**
**Dashed horizontal lines (green) indicate present-day temperatures in Secret Chamber and Whiterock Cave, respectively. Bottom: $\delta^{18}O_{cc}$ records from stalagmites SC02 (Buckingham et al., 2022) and WR5_B (Meckler et al., 2012). YD: Younger Dryas, ACR: Antarctic Cold Reversal, H-1: Heinrich Stadial 1.**

### 4.4 Polar amplification

The two Borneo temperature records from SC02 (Løland et al. 2022) and WR5_B (this study) show clear linear correlations
with Antarctic temperature anomalies both from Landais et al. (2021) and Jouzel et al. (2007), illustrated in Fig. 9a,b and
Fig. A7a,b, respectively, where the slope of the regression line reflects a site-specific polar amplification factor, i.e., the ratio



between Antarctic and Borneo temperature changes ($\Delta T_{Antarctic}$/$\Delta T_{Borneo}$). For simplicity, age and temperature uncertainties of the records were not considered for the fits. We found very strong correlations between Antarctic $\Delta T$ and the SC02 record ($R^2$ = 0.97) and slightly weaker correlations for the WR5_B record ($R^2$ = 0.87 and 0.89, respectively; see Table 2). The

weaker correlation of the WR5_B record likely relates to larger age uncertainties. The slopes of the regression lines differ insignificantly between the SC02 and WR5_B records, but they reflect the systematically higher glacial temperatures in the Antarctic $\Delta T$ reconstruction of Landais et al. (2021) compared to that of Jouzel et al. (2007). The intercepts of the regression lines at Antarctic $\Delta T$ = 0 closely accord with stalagmite-derived average Holocene temperatures from SC02 and WR_MC1. We note that SC02 temperatures used for the linear fits were corrected for the altitude difference between Secret Chamber

and Whiterock Cave (see Section 4.3). The strong correlation between the Borneo and the Antarctic records that were derived from different climate archives with independent temperature proxies can be taken as evidence for the reliability of these paleo-temperature reconstructions.

**Table 2: Correlations between Borneo temperature records ($T_{corr.}$) and Antarctic temperature anomalies $\Delta T$. The slopes of the**
**linear regression lines (Fig. 9a,b and Fig. A7a,b) represent empirical estimates of polar amplification factors while the respective intercepts at Antarctic $\Delta T$ = 0 are equivalent to Holocene Borneo temperatures estimates (both given with 95% confidence interval (CI)). Holocene temperatures (with 2SEM) derived from stalagmite SC02 and from the drill core sample\* (WR_MC1) collected in Whiterock Cave (Fig. S5) are shown in the rightmost column for comparison.**

|  | Jouzel et al. (2007) | | | Landais et al. (2021) | | | Holocene |
|---|---|---|---|---|---|---|---|
|  | slope | $T_{corr.}$ at $\Delta T$ = 0 (°C) | $R^2$ | slope | $T_{corr.}$ at $\Delta T$ = 0 (°C) | $R^2$ | reference (°C) |
| SC02 | 2.42 ±0.23 | 23.22 ±0.20 | 0.97 | 2.21 ±0.22 | 22.86 ±0.17 | 0.97 | 23.1 ± 0.28 |
| WR5_B | 2.34 ±0.25 | 23.09 ±0.25 | 0.89 | 2.13 ±0.24 | 22.87 ±0.24 | 0.87 | 23.3 ± 0.25* |


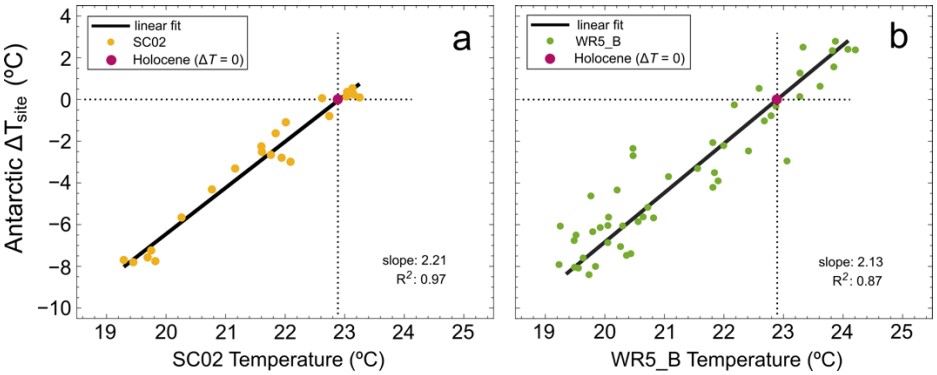

**Fig. 9: Correlations between Antarctic temperature anomalies $\Delta T_{site}$ (Landais et al., 2021) and a) altitude- and sea level corrected**
**SC02 temperatures (Løland et al., 2022) covering the time interval from 23-6 ka and b) sea level corrected WR5_B temperatures (this study) covering the time interval from 460-333 ka. The slopes of the regression lines represent polar amplification factors and the purple dots at $\Delta T_{site}$ = 0 are resulting predictions of average late Holocene Borneo temperatures.**



## 5. Conclusions

The present study provides a new record of land temperatures from tropical Northern Borneo derived from speleothem fluid
inclusions covering the time interval between 460 and 333 ka ago. Our results reinforce the great potential of NA microthermometry for obtaining highly precise and accurate paleo-temperature data, indicated by small standard errors and the consistency of temperature data across different stalagmites, and in comparison with temperature records from other archives.

Our record shows that Borneo land temperature evolved in step with changes in Southern Hemisphere (Antarctic)
temperature across T-V and T-IV, corroborating previous findings of Løland et al. (2022) observed for the last glacial termination (T-I). Just as during T-I, deglacial warming starts with the beginning of Heinrich stadials involving relative drying in Northern Borneo and cooling in the Northern Hemisphere. We thus confirm a widespread warming effect of the deglacial AMOC reduction, as well as a clear decoupling between temperature and hydroclimate in the West Pacific Warm Pool region during deglacial Heinrich events.

We find a strong linear correlation between Antarctic $\Delta T$ and the two Borneo records, SC02 (LGM to Holocene; Løland et al., 2022) and WR5_B (MIS 12 to MIS 9; this study), yielding statistically indistinguishable polar amplification factors, which raises the question whether polar amplification can, as a first order approximation, be assumed constant over the past 460 kyr. Further temperature records from Borneo stalagmites covering other time intervals at higher temporal resolution and with better age constraints than WR5_B will allow testing this hypothesis and investigating the sensitivity of tropical
land temperature to short-term millennial scale climate fluctuations.



## 6. Appendix A: Supplementary figures

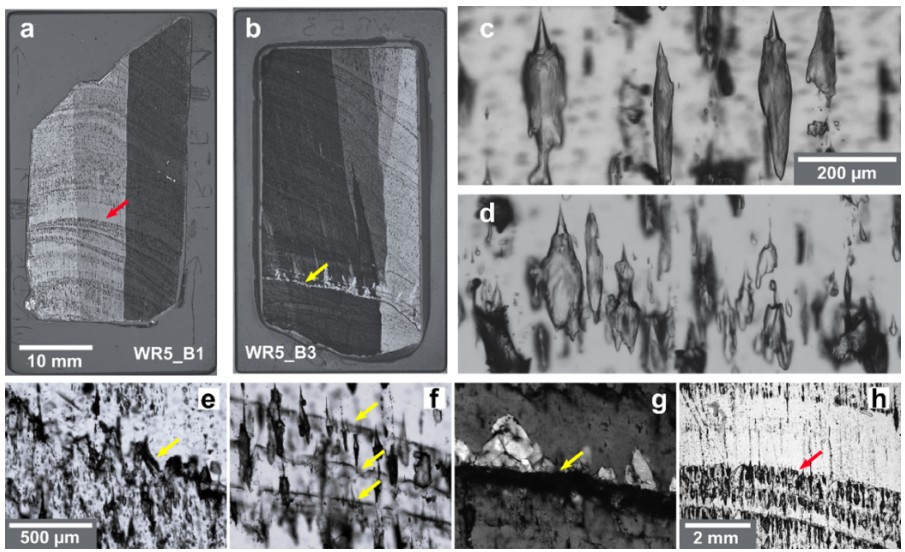

**Fig. A1: a, b)** Polished stalagmite thick sections (ca. 200 μm) viewed in transmitted light with partly crossed polarizers. The calcite
fabric is characterized by large composite crystals that are built up from columnar crystal units of nearly uniform *c*-axis
orientation. Arrows indicate the growth hiatuses shown in h) and g), respectively. **c, d)** monophase liquid fluid inclusions of
various shapes and sizes. The inclusions are typically located between adjacent columnar crystal units and are elongated parallel
to the calcite *c*-axis and growth direction, respectively. **e)** Hiatus with dissolution features (Type E surface). **f)** Hiatus with even
surface that is outlined with countless tiny sub-micron inclusions. **g)** Calcite crystals with random crystallographic orientation on
the hiatus surface shown in b). **h)** Enlarged image of the prominent hiatus at the beginning of T-IV shown in a). Sample positions 5
and 6 (Table 1) are right above and below the hiatus, respectively.

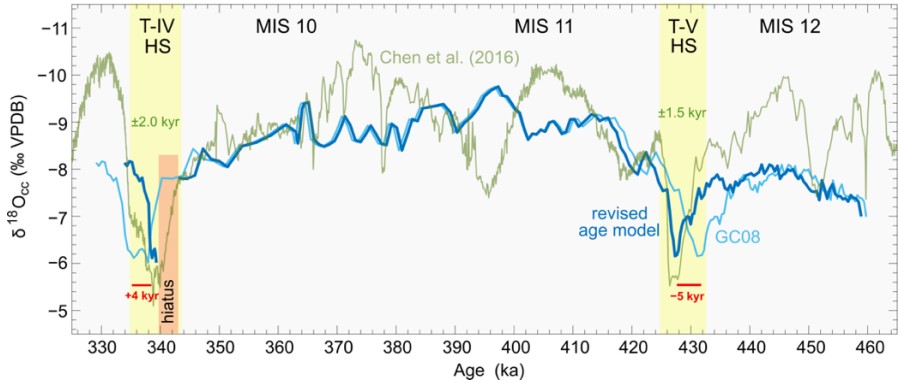

**Fig. A2:** Revision of the WR5_B age model using the calcite δ$^{18}$O record. Light blue: the original WR5_B age model (Meckler et
al., 2012) was obtained by matching the δ$^{18}$O$_{cc}$ record to the one from stalagmite GC08 (not shown). GC08 originates from Green
Cathedral Cave, about 10 km northeast of Whiterock Cave and exhibits a better age model, though still with large age
uncertainties in the order of ± 12 kyr. In the present study, we used the Chinese Monsoon record shown in green (Chen et al. 2016)
to improve the WR5_B age model at the two glacial terminations T-V and T-IV using the pronounced δ$^{18}$O$_{cc}$ excursions that are
related to Heinrich events and Northern Hemisphere cooling. Age errors of the monsoon record are about ±1.5 kyr and ±2.0 kyr
for the time intervals around T-V and T-IV, respectively. The revised WR5_B age model is shown in blue featuring a 3 kyr hiatus
at the beginning of T-IV (see Section 2.2). The revised model shifts the original ages around T-V and T-IV by ca. -5 and +4 kyr,
respectively. Note the reversed y-axis.



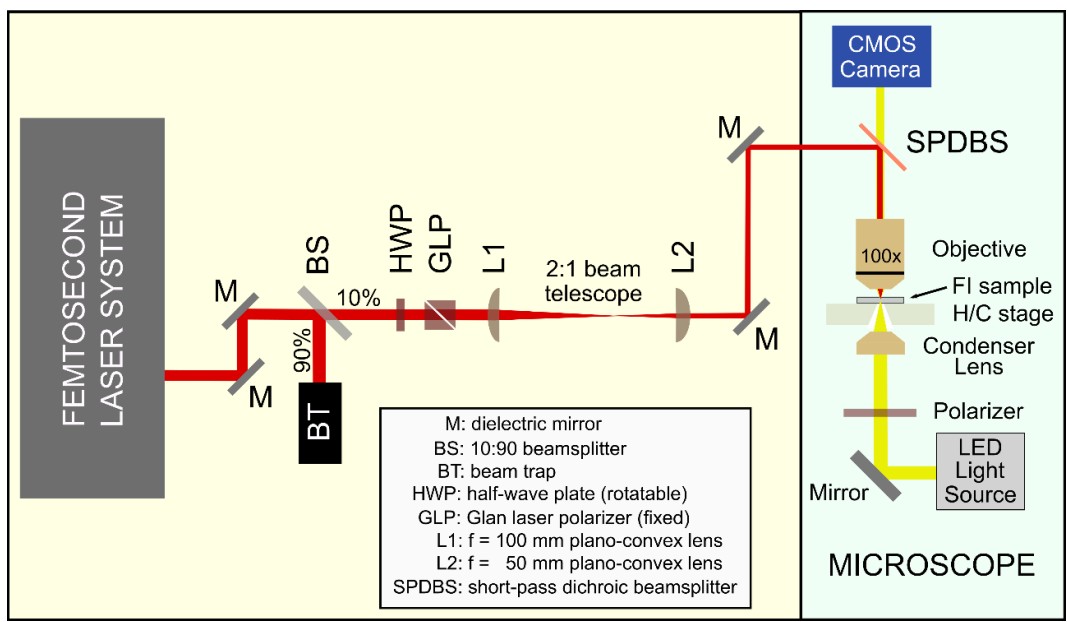

**Fig. A3: Schematic representation of the analytical setup used for NA microthermometry. The femtosecond laser system ((CPA-2101, Clark-MXR, Inc.) provides ultra-short laser pulses with a nominal pulse width of 150 fs (1 fs = 10⁻¹⁵ s) and an output power of ca. 0.9 W at 1kHz repetition rate. For NA microthermometry the output power is attenuated to ca. 20 mW and the laser is operated in single pulse mode. The laser wavelength is 775 nm and accordingly, NIR-coated optics are used to direct the laser beam to the microscope. Via two dielectric mirrors (M) the beam passes a 10:90 beamsplitter (BS), transmitting only 10 % of the laser light, while reflecting 90 % into a beam trap (BT). After this second attenuation step, the beam passes a rotatable half-wave plate (HWP) and a fixed Glan laser polariser (GLP). Rotation of the HWP by 45 ° changes the polarisation direction of the laser from horizontal to vertical and in combination with the fixed GLP the laser power can be further attenuated and fine-tuned. After the polariser the laser beam passes a 2:1 beam telescope consisting of two plano-convex lenses (L1 and L2) with focal lengths of 100 and 50 mm, respectively. The beam telescope reduces the beam diameter to ca. 3 mm and re-collimates the beam before it is coupled into the microscope light path by means of two more mirrors (M) and a short-pass dichroic beamsplitter (SPDBS, Chroma ZT720spxxr-uv-UF1) that is mounted in a dual port intermediate tube on the microscope. The dichroic beamsplitter reflects the 775 nm wavelength of the laser into the microscope objective and transmits the visible spectral range from the microscope illumination below ca. 700 nm. Kinematic mirror mounts are used to couple the laser beam properly into the microscope light path. The microscope (Olympus BX53) is equipped with a Linkam THMSG600 heating/cooling stage, an Olympus LMPLFLN 100x/0.8 long working distance objective (WD: 3.4 mm) and a digital monochrome camera (pco.edge 3.1). For microthermometry the 100x objective is placed into a metal jacket that fits through an open lid on the Linkam stage, sealed with rubber o-rings.**





**Fig. A4: WR5_B data plots from sample positions 1-6**





**Fig. A4: WR5_B data plots from sample positions 7-11**





**Fig. A4: WR5_B data plots from sample positions 12-17**



**Fig. A4: WR5_B data plots from sample positions 18-23**





**Fig. A4: WR5_B data plots from sample positions 24-29**



**Fig. A4: WR5_B data plots from sample positions 29-34**








**Fig. A4: WR5_B data plots from sample positions 35-40**



**Fig. A4: WR5_B data plots from sample positions 41-46**





**Fig. A4: WR5_B data plots from sample positions 47-49**

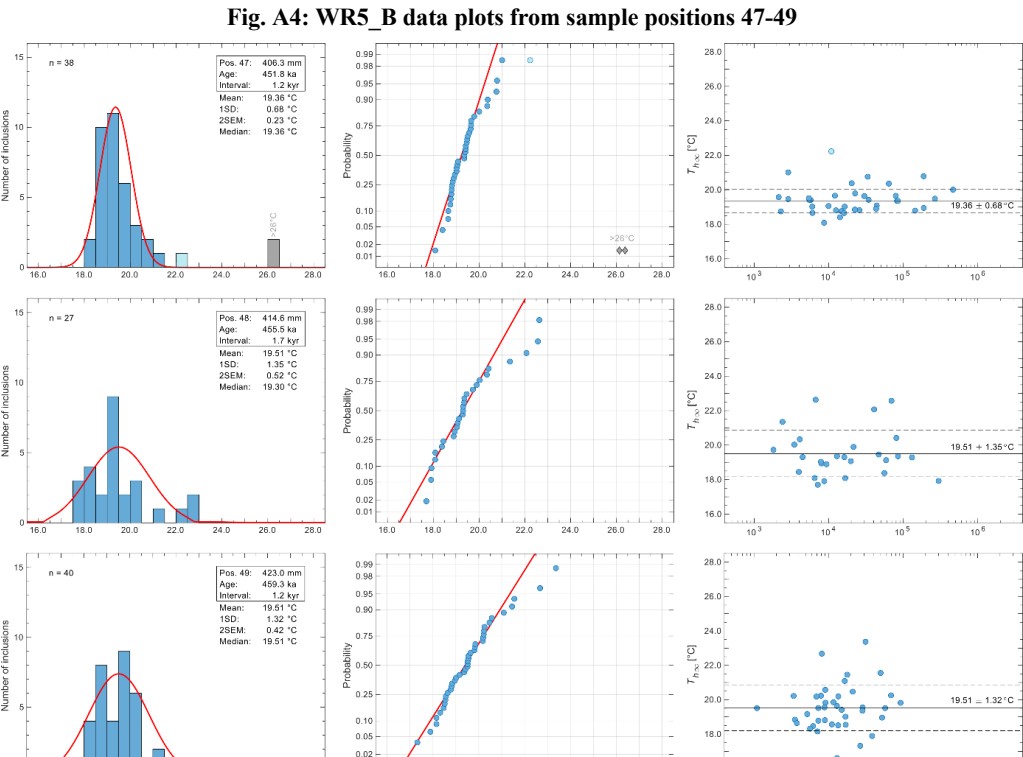


**Fig. A4: WR5_B data plots for sample positions 1-49. Left column: histogram plots of $T_{h\infty}$. Inclusions with $T_{h(obs)}$ greater than the threshold value are marked in grey, $T_{h\infty}$ values that were classified as outliers based on a 4MAD criterion are marked in light blue and were not considered for statistical analysis. Red curves are Gaussian fits to the data based on the mean and standard deviation (1SD) of the distribution; n = total number of inclusions, SEM = standard error of the mean. Centre column: cumulative**

**distribution plots of the $T_{h\infty}$ data. Note the straight red line represents the same Gaussian function as in the histogram plots but now plotted on a logarithmic probability scale. Right column: $T_{h\infty}$-$V$ plot illustrating the size range of the analysed inclusions and the scatter of the $T_{h\infty}$ values in the individal samples. Note, there is no correlation between $T_{h\infty}$ and inclusion volume, confirming that the observed scatter of $T_{h\infty}$ is volume independent. Grey line: mean; dashed black lines: ±1SD. Numerical values are reported in Table S2.**







**Fig. A5: Drill core sample WR_MC1 taken from an actively growing stalagmite covering the Late Holocene (bottom age: 3.45 ±0.25 ka). Upper panels: Left: image of the thick section in transmitted plane polarised light. Fluid inclusions were analysed at the three sample positions indicated on the image. Middle: same section viewed in cross-polarised light, showing the large domains of uniform *c*-axis orientation of the columnar calcite fabric. Right: histogram plots of the $T_{h\infty}$ distributions for the three sample positions. Note, for the top sample we used a 3MAD instead of the common 4MAD outlier criterion to get rid of the extreme values of the high-temperatures tail. Lower panels: cumulative distribution plots of the $T_{h\infty}$ data and $T_{h\infty}$-V diagrams. Numerical values are reported in Table S3.**



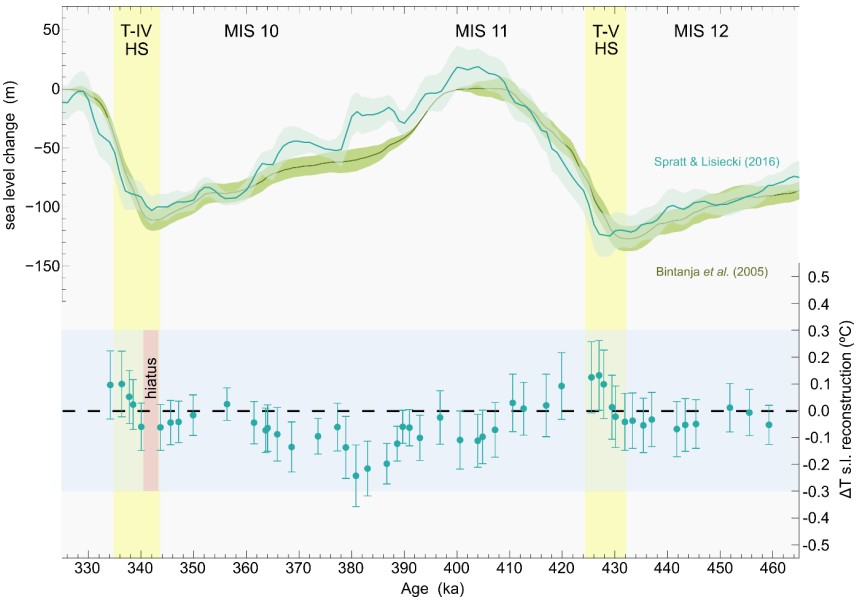


**Fig. A6.** Top: Comparison of sea level reconstructions from Bintanja (2005) and Spratt&Lisiecky (2016) plotted on the LR04 time scale (Lisiecki & Raymo, 2005). Note, the lag of sea level rise with respect to deglacial warming at the T-V and T-IV Heinrich stadials indicated with yellow bars. The lag is more pronounced in the record of Spratt&Lisiecky that, in addition, features more variability during the glacial inception from MIS 11 to MIS 10, which is in accordance with millennial scale climate variations observed in the Antarctic temperature and $CO_2$ records. Bottom: Differences in sea level corrected Borneo land temperatures $T_{corr}$ depending on the chosen sea level reconstruction (cf. Table S1). $\Delta T$ becomes largest at the Terminations and during the glacial inception with error bars relating to the uncertainty (1SD) of the sea level estimates. The variation of $\Delta T$ is within a bandwidth of ±0.3 °C (blue shaded area), and thus, within the average error (2SEM) of the reconstructed $T_{corr}$ values.


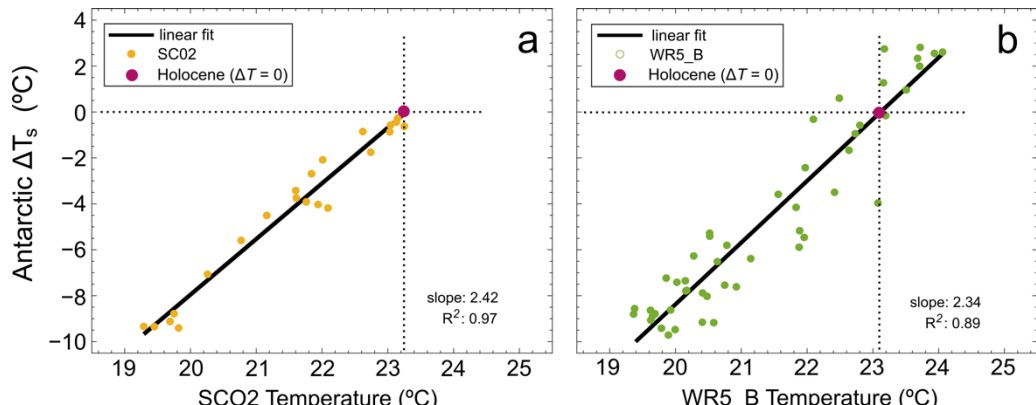

**Fig. A7:** Correlations between Antarctic temperature anomalies $\Delta T_s$ (Jouzel et al., 2007) and a) altitude- and sea-level corrected SC02 temperatures (Løland et al., 2022) covering the time interval from 23-6 ka and b) sea level corrected WR5_B temperatures (this study) covering the time interval from 460-333 ka. The slopes of the regression lines represent polar amplification factors and the purple dots at $\Delta T_s = 0$ are resulting predictions of average late Holocene Borneo temperatures.




**Data availability:** https://doi.org/10.5281/zenodo.14864527

**Author contribution:** YK and ANM: study design. YK: measurements, text, figures and tables; LP: statistical analysis, text
revision; AF: drip water reconstruction; KMC: sample collection, text revision; ANM: sample collection, cave monitoring,
text revision.

### Acknowledgements

This study was funded through the Norwegian Research Council (NFR-Grant 262353/F20 to A.N.M.) and the European

Research Council (ERC-Grant 101001957 to A.N.M.). K.M.C. acknowledges support through NSF Awards (1203785 and

1502830). We are deeply grateful for the expert guidance of Syria Lejau and Jenny Malang during fieldwork in Borneo, as

well as the support of the Gunung Mulu National Park management and staff. Jess Adkins is acknowledged for facilitating

and contributing to the original work on WR5 and other Northern Borneo speleothems. We thank Elena Rogmann for

analysing some of the samples for the WR5_B record during her internship at the Department of Earth Science, University of

Bergen.

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
