# Peer review of "Tropical temperature evolution across two glacial cycles derived from speleothem fluid inclusion microthermometry"

_EGUsphere, 2025_

## Author Comment (AC1)

Reply to reviewer comments:

Reviewer 1: Charlotte Honiat, 07 Apr 2025

COMMENT:

> Line 98 – 99 « Although the two temperature records do not cover the same time interval it is safe to assume that present-day cave temperature in Whiterock Cave closely reflects (multi) annual mean outside air temperature without seasonal bias. »
>
> Here the argument could benefit or be supported by a check of the climatological reanalysis data for the same time interval.

REPLY

> Following the reviewer's suggestion we studied the temperatures from the ERA5 reanalysis for the area around Mulu airport (3.85-4.25°N; 114.60-114.85°E) using the KNMI Climate Explorer (https://climexp.knmi.nl). To our surprise, the resulting mean temperature for the period from 2006 to 2012 (26.1 °C) was found to be significantly higher than the average of the measured Mulu airport temperatures for the same period (24.3 °C) that we intitially used as a reference surface air temperature. Although this reference outside temperature would fit very well with the average cave temperature of 23.7 °C measured between 2018 and 2020, when allowing for altitude effects and the warming trend in the region, the obvious discrepancy with the reanalysis data motivated us to revisit existing monitoring data of outside air temperatures recorded in the period 2018-2020 at two locations in Mulu National Park and at the entrance of Whiterock cave. These monitoring data were also significantly higher than cave temperature, with 25.5 °C and 26.6 °C at the two park locations and 25.1 °C at the cave entrance. Previously, we had attributed these warmer temperatures to local heating effects at the sites of logger deployments, however, it turns out that the monitoring data agree very well with the reanalysis data for the same period, yielding an average temperatures of 26.7 °C. Although we do not yet have an explanation for the lower temperatures at Mulu airport, we no longer think that these data are representative for the area.

> The average temparture measured at the cave entrance, which is consistent with the other logger data and the reanalysis data when corrected for the cave altitude, was found to be 1.4 °C higher than the air temperature inside the cave measured during the same time interval. This raises fundamental questions about the temporal relation between cave and surface air temperature and we find this topic important enough to warrant adding a new parapgraph 2.1 briefly discussing the monitoring data, the physical processes involved in the transfer of the outside temperature signal into the cave and their potential implications on the interpretation of cave temperature records.

> The new paragraph will read as follows:

**2.1 Temperature monitoring**

Stalagmite WR5 was collected in 2008 from Whiterock Cave (Gunung Mulu National Park, Northern Borneo, Malaysia; 4.1°N, 114.9°E) in a cave chamber about 350 m from "Midnight Entrance" (180 m a.s.l.). The site is ca. 100-140 m below the surface and present-day air temperature in the cave chamber is 23.7 °C. Continuous temperature monitoring close to the location of WR5 over a 30-month period, from March 2018 until September 2020, revealed minor fluctuations of the cave air temperature of about ±0.1 °C (total amplitude), indicating very stable temperature conditions throughout the year. For comparison, temperatures recorded at the cave entrance during the same period show daily and seasonal fluctuations and a average air temperature of 25.1 °C over a full 2-year cicle (July 2018-June 2020), which is 1.4 °C higher than the temperature recorded inside the cave. Surface air temperatures measured at two locations in Gunung Mulu National Park (at ca. 30 m a.s.l.) over the same 24-months period yield average temperatures of 25.5 (rainforest) and 26.6 °C (open clearing). These temperatures are consistent with the temperature at the cave entrance when allowing for the 150 m altitude difference with a lapse rate of around 0.6 °C/100 m (Løland et al., 2022). ERA5-T2m reanalyisis land temperatures (KNMI Climate Explorer, https://climexp.knmi.nl) for the nearby lowland area at 3.85-4.25°N, 114.60-114.85°E for the same period yield an average temperature of 26.7 °C, which is in agreement with our monitoring data of outside air temperatures (Fig. A1; see Appendix). We note that previously reported temperature data recorded between 2006 and 2012 at the nearby meteorological station at Mulu airport (24 m a.s.l.), in contrast, indicate significantly lower surface air temperatures, averaging at 24.3 °C over the entire 6-year period with annual mean temperatures varying between 24.0 and 24.7 ˚C. The respective ERA5-T2m average temperature over this period is 26.1 °C. Although the altitude-corrected 6-year average temperature from Mulu airport (23.4 °C) is close to the measured cave temperature and matches even better when considering a regional warming trend of about 0.3 °C/decade (based on reanalysis data), the airport temperature data are in clear contradiction with the other temperature records, and we therefore consider them as not representative for the area.

The observed deviation between cave and outside air temperature (as measured at the cave entrance) means that the assumption of identity of cave and (coeval) surface air temperature is not applicable to Whiterock Cave, and that their relationship is more complex. Although it is generally accepted that cave temperature is controlled by surface temperature, theoretical studies have demonstrated that the transfer of the temperature signal into the cave involves a systematic lag of the cave temperature in response to changes of outside surface air temperature: The heat flux into the cave is governed by different physical processes including heat conduction through the overlaying karst bedrock (Dominguez-Villar et al., 2020), convective heat transport by water infiltration (Badino, 2004, 2010) and advective heat transport related to air flow through the cave galleries (Luetscher & Jeannin, 2004). The relative contribution of these processes controls the heat flux, and thus, the lag time of the cave temperature relative to the outside temperature signal and is likely cave-specific, depending on the geometry and climatic setting of the cave. The delay of the signal transfer acts as a low-pass filter resulting in a potential attenuation of the amplitude of outside temperature anomalies in the cave (Dominguez-Villar et al., 2020). This is most obvious on daily and seasonal time scales but depending on the actual lag or equilibration time of the system, attenuation of the outside temperature signal may also occur on decadal, centennial or even longer time scales.

In a tropical setting like Northern Borneo with high annual precipitation of 4000-5000 mm/m$^2$, convective heat transport through fracture flow of infiltrating water is likely dominating the heat flux into the cave. Badino (2010) pointed out that the infiltrating rainwater is typically colder than mean surface air temperature and may decrease the cave temperature below outside air temperature, thus resulting in a systematic offset. If this holds true for Whiterock Cave, it would be a an additional factor contributing to the total temperature differential between cave and surface air. Given a rapid warming in the region by about 1.5 °C since the late 1960's according to the reanalysis data, a deviation of the cave temperature from outside air temperature is therefore not surprising, but is to be expected.

How does this affect the interpretation of the cave temperature record? The amplitude of a change in cave temperature can generally be considered as a minimum estimate of the actual amplitude of the surface air temperature signal. The lower the frequency, i.e. the longer the duration of the temperature anomaly relative to the signal delay, the closer the cave temperature approaches the amplitude of the surface signal. In the present study we are dealing with cave temperature changes on millennial and

orbital time scales and considering the climatic setting and depth of the cave below surface, we assume that the amplitude of the reconstructed temperature changes in Whiterock Cave closely reflects the amplitude of the outside temperature signal. We will get back to this in Sections 4 and 5.

In addition to paragraph 2.1, we will add a new figure (A1) in the Appendix, showing the monitoring and reanalysis data for the period 2018-2020:

[Figure]

**Fig. A1: Surface and cave air temperatures recorded between March 2018 and September 2020. Blue: Mulu Park head quarter (HQ), located in a forest clearance about 30 m a.s.l.; red: location along the tourist boardwalk in the tropical forest (30 m a.s.l.); purple: Whiterock "Midnight" cave entrance (180 m a.s.l.); black: ERA-5 T2m temperature reanalysis for the area 3.85-4.25°N, 114.60-114.85°E; green: Whiterock cave temperature. Average temperatures over a 2-year period (grey box) are shown on the right of the respective records. In addition, the reconstructed late Holocene reference temperature from the WR_MC1 drill core sample is indicated as well (brown).**

COMMENT:

Line 147 we additionally took *into* account

REPLY:

We will rephrase the sentence as follows

"At Termination IV, the new age model additionally discloses a major hiatus (red dashed line in Fig. 2) that stands out owing to a clear discontinuity in the isotope $\delta^{18}O_{cc}$ record."

COMMENT:

And « major hiatus (red dashed) » : is it the growth hiatus shown in Fig A1 ? If yes it would be good to refer to the figure ;

REPLY:

Here we refer to Fig. 2 where the red dashed line is shown. In the caption of Fig. 2 we refer further to Fig. A1 a) and h), and will now also refer to Fig A2 and Fig. 5b-d.

In the caption of Fig. A1 we will add a reference to Fig. 2.

COMMENT:

Figure 4 : is this inclusion from the studied stalagmite ? could you mention from which stalagmite it comes from ?

REPLY:

Yes, the inclusion is from stalagmite WR5_B and was originally analysed for the study of Meckler et al. (2015). We will add this information to the figure caption as follows:

**"Image sequence illustrating the different states (a-e) of a WR5_B fluid inclusion during microthermometric measurements. …"**

COMMENT:

Line 320 : you mention the lapse rate uncertainty, could you give number ?

REPLY:

The lapse rate uncertainty (±0.1 °C) is mentioned just above in line 317 toghether with a reference to Løland et al. (2022). We therefore think it is not necessary to mention it again in line 320.

COMMENT:

Line 349 : « we employed our temperature record to calculate the oxygen isotopic composition of the calcite supplying drip water d18Odw from which the stalagmite formed » I think this sentence need to be re-written ; isn't the drip water supplying speleothem formation and not the calcite ?

REPLY:

We will rephrase the sentence to avoid potential confusion:

« we employed our temperature record to calculate the oxygen isotopic composition of the drip water $\delta^{18}O_{dw}$ from which the stalagmite-forming calcite precipitated »

COMMENT:

Line 351 : Why did you use the empirical speleothem calibration for the calcite-water oxygen isotope fractionation proposed by Tremaine et al. (2011) ; and not other equations like Kim and O'neill (1997) ? I would be interesting to see the difference between the most commonly used equation in the litterature. Maybe this choice is based on modern monitoring data, if so, it would be worth mentionning it.

REPLY:

Meckler et al. (2015) have shown that temperatures calculated from measured $\delta^{18}O_{FI}$ and $\delta^{18}O_{cc}$ using the calibration of Kim and O'Neill (1997) are unrealistically low,

whereas the empirical speleothem calibration of Tremaine et al. (2011) performed much better, yielding reasonable temperature values. The study of Meckler et al. (2015) that was conducted on WR5_B is cited. While we realise that open questions still remain regarding the different calcite-water oxygen isotope fractionation equations, we feel that such a discussion is beyond the scope of our study.

COMMENT:

Line 371 : « the previous interpretation of relative drying » not sure what « previous » refers to here ? Maybe a reference is needed

REPLY:

"previous interpretation" refers to the first sentence of the Section (lines 352-354), in which we refer to the paper of Meckler et al. (2012). For clarity, we will cite the paper here anew.

COMMENT:

Line 392 : That is correct but it would be good to give estimates of uncertaities on both record

REPLY:

We will add age uncertainties of the Borneo and Antarctic temperature records for the T-V time interval (±1.5 and ±4.0 kyr, respectively)

COMMENT:

Line 459 : Give altitude difference in paranthesis

REPLY:

Will be done

COMMENT:

Figure 8 : You discuss above (between line 459 and 469) that you corrected for the difference in cave altitude between the Secret Chamber cave (Loland 2022) and Whiterock Cave (this study) so why not plotting a corrected version to make comparison easier for the reader ?

REPLY:

In Fig. 8 we decided to plot the original SC02 data from Løland et al. (2022) to illustrate the necessity of altitude corrections for comparing stalagmite records from different caves, in part because this allows direct comparison with the measured present-day cave temperatures which we would like to keep uncorrected to allow the reader to observe the offset between the caves today. We have considered plotting the altitude corrected SC02 record, but we doubt that this would make a visual

comparison of the two records much easier and see more disadvantages than advantages in doing so.

COMMENT:

Why termination T-IV is not compared with T-I and T-V here ? I understand there is a hiatus during part of the glacial-interglacial transition but this hiatus also holds information ; is the timing of the hiatus comparable to an HS event-like ? A hiatus would fit with your argument of relative drying during Heinrich Stadials

REPLY:

We compared T-I to T-V because both terminations are well resolved in our records and precede interglacials (Holocene and MIS11, respectively) that are characterised by relatively weak insolation forcing, which makes a comparison of the two terminations particularly interesting. Moreover, we used the comparison of T-V and T-I to evaluate the consistency of our temperature reconstructions across different stalagmites and caves.

Indeed, we refrained from comparing T-IV to T-I because of the hiatus and because we suspect that we may have missed the peak temperature at the end of the Heinrich stadial. We agree that the hiatus at T-IV fits at first sight with the argument of relative drying during the Heinrich stadial, however, a closer look at the $\delta^{18}O_{cc}$ record (Fig. 5 and A2) indicates that the timing of the hiatus does not coincide with the maximum of the $\delta^{18}O_{cc}$ excursion (note the reverse axis) but it starts already at the beginning of the Heinrich stadial, while stalagmite growth continues again at or right after the $\delta^{18}O_{cc}$ maximum. We assume that the hiatus at T-IV might be rather related to cave-specific hydrology than reduced rainfall during T-IV. A relative drying in Borneo is furthermore not expected to result in cessation of dripping and speleothem growth, given the tropical location of the site. Finally, we note that T-V does not exhibit a prominent hiatus like T-IV although the excursion of $\delta^{18}O_{cc}$ associated with the Heinrich stadial is of similar amplitude.

COMMENT:

Figure 9 : How did you select individual datapoint ? Is it based on each individual FI measured datapoint from your record and then the corresponding T° from the Antarctic record ? Or is at a point at a regular time interval from your interpolated record ? Could you clarify in the figure caption ?

REPLY:

The data points are reconstructed temperatures $T_{corr}$ plotted against Antarctic temperatures at the same ages. However, since the Antarctic record has a higher temporal resolution, we employed a Gaussian filter with bandwdith (1SD) of 0.2 kyr to obtain representative Antarctic temperature values. We will revise the figure caption as follows:

**Fig. 10: Correlations between Antarctic temperature anomalies D$T_{site}$ (Landais et al., 2021) and a) altitude- and sea level corrected cave temperature data ($T_{corr}$, n = 22) derived from stalagmite SC02 (Løland et al., 2022) covering the**

**time interval from 23-6 ka and b) sea level corrected cave temperatures ($T_{corr}$, n = 49) derived from stalagmite WR5_B (this study) covering the time interval from 460-333 ka. The slopes of the regression lines represent polar (Antarctic) amplification factors and the purple dots at D$T_{site}$ = 0 are resulting predictions of average late Holocene Borneo temperatures. Due to the higher temporal resolution of the Antarctic record, we employed a Gaussian filter with a bandwidth (1SD) of 0.2 kyr to obtain representative Antarctic temperature values.**

COMMENT:

> Figure A4 : There are two different plot with Sample Pos. 29 (page 28 and 29)

REPLY:

> This observation is correct, but this is intentional. At sample positions 9 and 29 we provide both a unimodal and a bimodal fit to the $T_{h\infty}$ data. We will add the following sentence to the figure caption:
>
> "At positions 9 and 29 the $T_{h\infty}$ distributions were considered as bimodal based on statistical tests. For comparison, we also provide the respective unimodal fits to the data."

COMMENT:

> Stalagmite WR5_B has previously been studied in a comparison study of different speleothem paleo thermometers (Meckler et al., 2015). I saw that the FI stable isotopes are plotted in figure 5 ; but what about the previous FI microthermometry data ? How does this newly produced data compare to the one in the study of Meckler et al. (2015) ? It could be briefly mentionned in the result section if uncertainty improvement have been made since that study, or shown in a supplementary figure in the appendix (showing the 2015 and 2025 data for the time period they overlap).

REPLY:

> Following the reviewer's suggestion, we will provide an additional figure in the Appendix, comparing the $T_{cave}$ data published in Meckler et al. (2015) with the $T_{cave}$ record of the present study. To do so, we reassessed the exact sample positions of the previous data points to check them for consistency with the present record. In the new figure we will show the $T_{cave}$ values of Meckler et al. (2015) both for the corrected as well as on the original sample positions and based on the revised age model.

[Figure]

**Fig. A6: Comparison of WR_5B temperatures ($T_{cave}$) from this study (blue) with previously published data (red; Meckler et al., 2015). To compare the two data sets, we reassessed the exact sample positions of the previous data points to check them for consistency with the present record (*cf.* Section 3.3). For some samples, deviations of the**

**originally assigned sample depths were found to be up to 4 mm. Sample ages are based on the revised age model and in the diagram, we show them for both the original (red) and corrected (brown) sample depths.**

Based on the reassessment of the sample positions, we will add $\delta^{18}O_{FI}$ data points with corrected ages to Fig. 5d and in Section 3.3.we will add the following text:

"….The ages of the $\delta^{18}O_{FI}$ data points were adjusted according to the revised WR5_B age model.

Moreover, a re-examination of the exact sample positions relative to the reference isotope transect revealed, for some of the samples, discrepancies of up 4 mm from the originally assigned sample depths. In Fig. 5d the $\delta^{18}O_{FI}$ data were plotted for both the original (green dots) and the corrected sample depths (brown dots) to illustrate the resulting age shifts. We note that this reassessment of the sample positions does not affect the oxygen isotope derived temperature estimates published in Meckler et al. (2015) because the corresponding d$^{18}O_{cc}$ values were not taken from the reference isotope transect but were measured on the crushed calcite powders remaining after fluid inclusion water isotope analysis."

The revised Fig. 5d and figure caption will look like this:

[Figure]

**Fig. 6: a) Sea level reconstruction with 1SD (Bintanja *et al.*, 2006), plotted on the LR04 time scale (Lisiecki & Raymo, 2005). Note the lag of sea level rise relative to temperature at the terminations. Yellow bars indicate T-IV and T-V Heinrich stadials. b) WR5_B temperature record (this study). Blue dots represent actual cave temperatures ($T_{cave}$) and green dots are the respective sea level corrected cave temperatures ($T_{corr.}$). For graphical reasons, error bars (2SEM) are indicated for $T_{corr.}$ only. Grey boxes indicate the $T_{corr}$ data that were used for calculating glacial and interglacial mean temperatures. c) WR5_B oxygen isotope record ($\delta^{18}O_{cc}$) of the stalagmite calcite (Meckler *et al.*, 2012), d) reconstructed drip water isotope record $\delta^{18}O_{dw}$ (small blue dots) derived from $T_{cave}$ and $\delta^{18}O_{cc}$ using the empirical calibration for oxygen isotope fractionation in speleothems of Tremaine et al. (2011) plotted with 95% CI (light brown band). Green dots are measured $\delta^{18}O_{FI}$ values from fluid inclusion water analysis based on the originally assigned sample depths (Meckler *et al.*, 2015, data from Bern lab) with error bars indicating 1SD of 2-3 replicate measurements. Brown dots result from the re-examination of the sample positions. The purple dashed line with the grey 95% CI band shows the ice volume correction that shifts glacial $\delta^{18}O_{dw}$ to more negative values. Note, in both c) and d) $\delta^{18}O$ is plotted on reverse axes.**

COMMENT:

General comment : The notation of all temperature have to be made homogeneous
(either space or no space between number and ±)

REPLY:

We will make the notation consistent throughout the text in this form: 22.16 ±0.32 °C.

---

## Author Comment (AC2)

Reviewer 2: Anonymous Referee, 01 May 2025

COMMENT:

Typo on Fig. A2: Cheng et al. 2016

REPLY:

This will be corrected both in the figure and the caption

COMMENT:

The authors may consider moving Fig. A2 to section 2.2, since many of the millennial-scale climate interpretations hinge on the detailed wiggle matching to Cheng et al. 2016

REPLY:

Following the reviewers suggestion, we will move Fig. A2 to section 2.2

COMMENT:

Section 3.2 left me wondering how the globally averaged sea level reconstructions from Bintanja et al. compare to regional sea level reconstructions off the coast of Vietnam, New Guinea, etc. Do small differences in regional vs global sea level at, for example, the LGM yield significant differences in temperature corrections? If so, how can these differences be appropriately accounted for in the $T_{corr}$ uncertainties deeper in time?

REPLY:

In Fig. A6 we compare the global sea level reconstruction of Bintanja et al. (2006) with that of Spratt&Lisiecki (2016). The two records show significant differences of up to 40 metres. In the supplementary Table S1 we provide $T_{corr}$ calculated for both sea level reconstructions and the resulting temperature differences $\Delta T_{corr}$ are plotted in Fig. A6. For the investigated time interval $\Delta T_{corr}$ values range from +0.13 to -0.24 °C, and thus, are within 2SEM of the reconstructed $T_{corr}$ values. Given this result of our sensitivity test, we can ascertain that small differences in regional vs. global sea level would not have a significant effect on $T_{corr}$.

COMMENT:

Although not my specialty, the term *Polar Amplification* is typically used to describe the enhanced polar temperature response relative to the tropics. This study presents data from the Western Pacific Warm Pool alone, which may not be representative of the broader tropical average. Clarifying this distinction will help avoid overgeneralization. The authors may also consider using the term *Antarctic Amplification* as opposed to polar.

REPLY:

We will specify more clearly what the term "polar amplification" refers to in this context and we will follow the reviewers suggestion to use either the term "Antarctic amplification" or add the term "Antarctic" in parentheses, i.e., "polar (Antractic) amplification.